# Plug-and-Play Spiking Operators: Breaking the Nonlinearity Bottleneck in Spiking Transformers

Xinzhe Yuan [1]   Xiang Peng [1]   Bin Gu [2]   Huan Xiong [1]

## Abstract

ANN-to-SNN conversion offers a practical, training-free route to spiking large language models. However, current pipelines primarily focus on spike-driven realizations for Transformer linear-algebra operations, while providing limited support for key nonlinear operators. This gap limits compatibility with neuromorphic-style execution constraints, where such nonlinearities typically require division, exponentiation, or norm computations that are not naturally supported by standard leaky integrate-and-fire dynamics. To solve this problem, we propose a plug-and-play framework that implements spike-friendly approximations for Transformer nonlinearities and integrates into existing ANN-to-SNN pipelines. Our method decomposes these nonlinear computations into three recurring primitives—division, exponentiation, and $\ell_2$ norms—and realizes them via population computation using LIF neuron groups, combined with lightweight bit-shift scaling to avoid floating-point arithmetic. By composing these primitives as modular operator blocks, our framework supports common Transformer nonlinearities (e.g., Softmax, SiLU, and normalization) without any fine-tuning. Experiments on a range of LLMs Transformers show that selectively replacing the targeted nonlinear operators incurs less than a $1\%$ accuracy drop across all evaluated tasks.

## 1. Introduction

Recently, spiking neural networks (SNNs) have gained increasing attention for energy-efficient, event-driven computation on neuromorphic hardware (Roy et al., 2019; Davies et al., 2018; Akopyan et al., 2015). These models have shown promising results in both computer vision and natural language processing (Cao et al., 2015; Zhou et al., 2023; Lv et al., 2023; Zhu et al., 2023). Meanwhile, Transformer-based foundation models, particularly large language models (LLMs), have become the dominant inference workloads. Their costs are largely driven by dense linear algebra and associated memory traffic (Horowitz, 2014).

Motivated by this, recent spiking Transformers and SNN-LLM efforts have increasingly adopted ANN-to-SNN conversion to reduce inference cost without expensive retraining (Rueckauer et al., 2017; Chen et al., 2025a;b; You et al., 2024). These methods primarily convert Transformer linear computations, such as attention matrix multiplications and feed-forward projections, into spike-driven implementations that leverage event sparsity for improved energy efficiency (Rueckauer et al., 2017; Yan et al., 2024).

**However**, this spike-centric research paradigm remains incomplete for Transformer architectures, as existing works rarely provide spike-based realizations of key nonlinear operators such as activation functions, normalization layers, and Softmax (Zhou et al., 2023; Shi et al., 2024). Although these components are often considered secondary from an energy-accounting perspective (Horowitz, 2014), they become decisive under strict spike-only deployment. They are difficult to represent with standard leaky integrate-and-fire (LIF) neurons, since they commonly require division or norm computations that do not naturally arise from LIF dynamics. More critically, on neuromorphic hardware platforms that support only spike-based primitives, continuous-valued states are unavailable, making conventional implementations a fundamental obstacle to deployment (Davies et al., 2018; 2021). Consequently, existing ANN-to-SNN approaches often fall short of truly end-to-end spiking Transformers under strict spike-only constraints, *including their nonlinearities* (Zhu et al., 2023; Lv et al., 2023).

To address this limitation, a few works approximate Transformer nonlinearities with spiking computation. But they typically require additional training, limiting their compatibility with standard ANN-to-SNN conversion pipelines (Tang et al., 2025). Motivated by these observations, we ask the following natural and important ques-

[1]IASM, Harbin Institute of Technology, China [2]School of Artificial Intelligence, Jilin University, China. Correspondence to: Bin Gu <gubin@jlu.edu.cn>, Huan Xiong <huan.xiong.math@gmail.com>.

*Proceedings of the $43^{rd}$ International Conference on Machine Learning*, Seoul, South Korea. PMLR 306, 2026. Copyright 2026 by the author(s).

tion:

> *Can we design plug-and-play spike-based conversion for nonlinear operators in ANN-to-SNN pipelines?*

In this work, we answer this question affirmatively by developing *training-free* spike-based replacements that are compatible with standard LIF dynamics. We identify three primitive nonlinear computations that repeatedly arise in Transformer inference: division, exponentiation, and $\ell_2$ norms, which form the computational core of Softmax, SiLU, and RMSNorm. Our spike-friendly realizations are built from population computation using LIF neuron groups, together with simple shift-based scaling that avoids floating-point arithmetic. By constructing approximations for these primitives and composing them as modular operator blocks, we obtain fully spiking realizations of the above Transformer nonlineariti es under strict spike-only constraints, without any additional fine-tuning.

These operator-level modules are composable and can be plugged into existing ANN-to-SNN conversion pipelines to enable end-to-end spiking Transformer blocks, while respecting the spike-only primitives and lightweight digital operations natively supported by neuromorphic hardware. Our main contributions are summarized as follows:

- **We propose a spike-friendly approach to approximate Transformer nonlinear operators.** Our method operates at the operator level and can be seamlessly integrated into existing ANN-to-SNN conversion pipelines without requiring any modifications to model weights.

- **We provide theoretical guarantees on the conversion error.** We show that our spike-based approximations of nonlinear operators admit provably bounded conversion errors under mild conditions. Additionally, we identify specific configurations that can achieve high approximation accuracy.

- **We empirically demonstrate the applicability of the proposed method across different models.** We test our method on two widely used ANN-to-SNN conversion frameworks. Additionally, we apply our approach to well-known models that have not previously been converted to SNNs, such as Qwen3, by selectively replacing their nonlinear operators while keeping the other computational parts unchanged, to verify the broad applicability of our method.

## 2. Related Work

Early ANN-to-SNN conversion methods approximate continuous ANN activations using firing-rate coding and scale

alignment. Diehl et al. introduce weight and threshold balancing to enable fast and accurate inference in deep spiking networks (Diehl et al., 2015). Sengupta et al. further extend this conversion paradigm to deeper architectures, demonstrating that VGG and residual networks can be converted to SNNs without retraining while maintaining competitive performance (Sengupta et al., 2019). As Transformers became the dominant model class, You et al. propose SpikeZIP-TF, which systematically applies ANN-to-SNN conversion to Transformer architectures by spiking the linear projections in attention and feed-forward layers (You et al., 2024). At the scale of large language models, Xing et al. introduce SpikeLLM, which constructs large spiking language models using saliency-driven spike allocation and reports efficiency–accuracy trade-offs compared with standard low-bit inference pipelines (Xing et al., 2025).

Distinct from the above works that primarily focus on dense linear algebra, Sorbet constructs spike-based realizations of nonlinear functions using shift-based discrete operations, and applies knowledge distillation and fine-tuning to align the spiking model with the original BERT behavior (Tang et al., 2025). Although it identifies the issue of non-linear operators, its method requires training and is incompatible with other pipelines.

## 3. Preliminary

**Spiking Neurons and Temporal Accumulation**    SNNs process information over discrete timesteps $t = 1, \ldots, T$. At each timestep, neurons emit binary spikes and communicate through temporally accumulated activity. As a result, real-valued quantities can be represented implicitly by spike counts over a finite temporal window.

In practice, ANN-to-SNN conversion methods typically rely on rate-based representations, where the accumulated spike activity approximates the activation of an artificial neuron:

$$a_{\text{SNN}} \approx \sum_{t=1}^{T} s^t \cdot \theta, \tag{1}$$

where $s^t \in \{0, 1\}$ denotes the spike at timestep $t$, and $\theta$ is the firing threshold. Under appropriate scaling, the expected spike count matches the quantized ANN activation, enabling a direct correspondence between ANN activations and SNN spike statistics (Rueckauer et al., 2017).

**Leaky Integrate-and-Fire Neuron**    LIF neuron is the most commonly used neuron model in SNNs. In discrete time, its dynamics are given by

$$v(t) = \lambda v(t-1) + I(t), \tag{2}$$
$$s(t) = \mathbb{I}\left[v(t) \geq \theta\right], \tag{3}$$
$$v(t) \leftarrow v(t) - s(t)\theta, \tag{4}$$

where $v(t)$ denotes the membrane potential, $I(t)$ the input current, $\lambda \in (0, 1]$ the leak factor, and $\theta$ the firing threshold, $\mathbb{I}[\cdot]$ denotes the indicator function.

This accumulate–threshold–reset mechanism allows LIF neurons to approximate linear transformations through temporal spike accumulation, forming the basis of most existing ANN-to-SNN conversion methods.

## 4. Method

LLMs heavily rely on certain nonlinear operations that involve exponentials, divisions, and square roots. Specifically, the following are the formulas for three key functions we focus on, which involve the ratio of a numerator and a denominator:

$$\phi_{\text{Softmax}}(x_i) = \frac{e^{x_i}}{\sum_j e^{x_j}},$$

$$\phi_{\text{SiLU}}(x) = \frac{x}{1 + e^{-x}}, \quad (5)$$

$$\phi_{\text{RMSNorm}}(x) = \frac{x}{\sqrt{\frac{1}{d}\sum_{i=1}^{d} x_i^2 + \epsilon}}.$$

We specifically focus on RMSNorm because it involves the most challenging part of normalization, the computation of the $\ell_2$-norm (via squaring, summation, and square root operations) followed by a division. The approximation of RMSNorm can also be easily extended to LayerNorm by using mean subtraction, which can be implemented through addition operations.

**Remark 4.1** (The Division Family). These normalization operations share a profound algebraic structure: they are inherently *fractional*—a signed numerator normalized by a strictly positive denominator. This observation suggests a modular spiking implementation strategy: rather than approximating the entire nonlinear function end-to-end, we decompose it into numerator and denominator components, each amenable to spike-friendly approximation. Crucially, we propose treating division itself as a *primitive operation in the spiking domain*, realized through a dedicated *division neuron ensemble* that emits spikes to represent the quotient. This approach not only preserves the structural fidelity of the original computation but also opens a new pathway toward building spiking neural networks with explicit awareness of underlying algebraic forms.

### 4.1. Core Building Blocks

#### 4.1.1. DIVISION NEURON GROUP

We start with the core division neuron group. Although the division operator is hardly computable by neurons, integer division serves as a promising alternative. By merely controlling the error of integer division within a reasonable

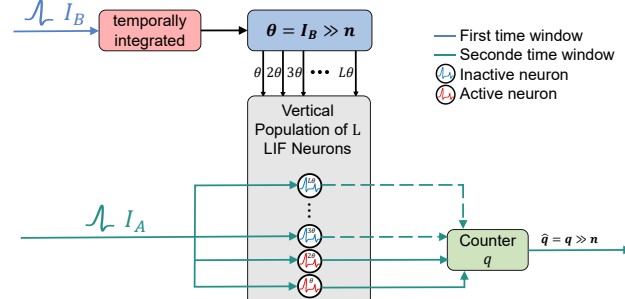

*Figure 1.* Overview of division neuron group.

range, it can be regarded as an approximate division operation. Following this rationale, we implement a spike-native division approximation using a population of $L$ standard LIF neurons with ordered thresholds and $\lambda = 1$.

As shown in Figure 1, in typical usage, both $I_A$ and $I_B$ are spike-coded signals. The division operation is carried out in a two-stage manner. In the first temporal window, the denominator input $I_B$ is temporally integrated to estimate a normalization scale. This scale is then held fixed and applied as population thresholds during a second temporal window, in which the numerator input $I_A$ drives the division neuron group. This separation reflects the fact that division depends on the aggregate magnitude of the denominator rather than its precise spike timing.

**Threshold construction.** Let $I_B(t)$ denote the spike-coded denominator input over the first temporal window of length $T$. We define the accumulated denominator

$$I_B \triangleq \sum_{t=1}^{T} I_B(t), \quad (6)$$

which yields a scalar quantity through temporal integration. From this accumulated value, we derive a base threshold via a right shift

$$\theta \triangleq I_B \gg n = \left\lfloor \frac{I_B}{2^n} \right\rfloor, \quad (7)$$

and assign the $i$-th neuron in the division population the threshold

$$\theta_i = i\,\theta, \qquad i = 1, \dots, L. \quad (8)$$

We choose both the temporal length $T$ and the population size $L$ as powers of two, and set

$$n = \log_2(TL), \quad (9)$$

so that the effective scale of $\theta$ matches the dynamic range of temporal accumulation. Once constructed, the thresholds $\{\theta_i\}$ remain fixed throughout the subsequent computation window.

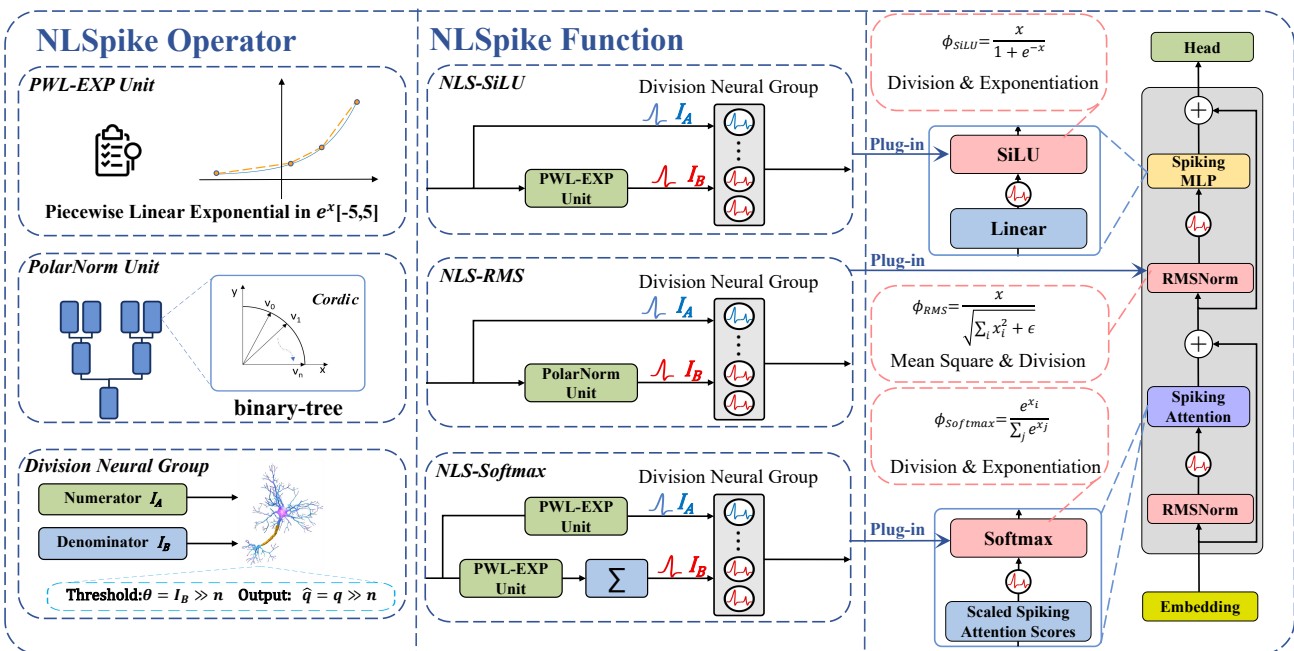

*Figure 2.* **Overview of the NLSpiking.** Each spike-unfriendly function (SiLU, Softmax, RMSNorm) is approximated using modular spiking Blocks: the Piecewise Linear Exponential (PWL-EXP) Unit, PolarNorm Unit, and Division Neuron.

**Population decoding as integer division.** During the second temporal window of length $T$, the spike-coded numerator input $I_A(t)$ is applied to the division neuron group. Neuron $i$ fires if and only if $I_A(t) \geq \theta_i = i\,\theta$. We decode the quotient by counting the number of active neurons,

$$q \triangleq \sum_{i=1}^{L} s_i, \; s_i \in \{0,1\}; \qquad \hat{q} = q \gg n, \quad (10)$$

which yields

$$\hat{q} = \sum_{t=1}^{T} \max\{i \mid v(t) \geq i\theta\} = \left\lfloor \frac{\sum_{t=1}^{T} v(t)}{\theta} \right\rfloor. \quad (11)$$

Noticing that $\left\lfloor \frac{\sum_{t=1}^{T} v(t)}{\theta} \right\rfloor = \left\lfloor \frac{\sum_{t=1}^{T} I_A(t)}{\theta} \right\rfloor$, substituting $\theta = I_B \gg n$ from (7) completes a spike-native discretization of division, in which the denominator is derived from a temporally integrated spike signal using only a shift operation.

**Remark 4.2** (Interpretation as Population Competition). The proposed division neuron group implements integer division by decomposing it into two stages: temporal estimation of a normalization scale from the denominator, followed by population-based competition under a shared input drive. The resulting winner index, or equivalently the number of active neurons, encodes the quotient. This mechanism relies solely on standard LIF dynamics and ordered thresholds, and does not require multi-level spikes or specialized neuron models.

### 4.1.2. POLARNORM UNIT (PN UNIT).

Having approximated the division, the second challenge arises from the norm, as square and square root operations are equally difficult to perform within spiking computations. To solve this problem, we introduce the *PolarNorm Unit (PN Unit)*. Specifically, we consider the expression $\sqrt{\sum_{i=1}^{d} x_i^2 + \epsilon d}$ and aim to approximate it using only additions, subtractions, comparisons, and bit-shifts—without general-purpose multiplications or square roots. From a geometric perspective, this term corresponds to the Euclidean norm of a vector, suggesting that the problem can be reformulated as a sequence of length-preserving vector reductions rather than explicit arithmetic operations.

Inspired by this, let $\mathbf{v} = [x_1, x_2, \ldots, x_d, \sqrt{\epsilon d}]$ be the augmented input vector. Our goal is to estimate $\|\mathbf{v}\|_2$ via a recursive reduction process using the *CORDIC-Hypot* algorithm (Volder, 1959), which approximates $\sqrt{x^2 + y^2}$ with only shift-add logic.

The approximation is performed using a binary-tree structure: adjacent elements of $\mathbf{v}$ are recursively merged via CORDIC-Hypot operations. Each CORDIC step updates $(x_k, y_k)$ as

$$x_{k+1} = x_k + d_k \cdot \frac{y_k}{2^k}, \quad y_{k+1} = y_k - d_k \cdot \frac{x_k}{2^k}, \quad (12)$$

where $d_k = \text{sign}(y_k)$. After $n$ iterations, the output $x_n$ approximates $\sqrt{x^2 + y^2}$ up to a constant gain. Applying this recursively over the tree yields a scalar norm estimate.

The final result is rescaled by a fixed inverse gain $1/K_n$ to correct for CORDIC accumulation. This constant factor is an integer power of 2, and therefore does not violate spike-friendly constraints: it can be absorbed into the scale $theta$ or approximated using fixed-point scaling.

### 4.1.3. PIECEWISE LINEAR EXPONENTIAL UNIT (PWL-EXP UNIT)

Last but not least, the computation of exp(x) remains infeasible in the realm of spiking computation. To address this, we introduce the PWL-Exp Unit.

We restrict our approximation to the interval $[-L, L]$, which covers the effective input range for most normalized activations. This range is divided into K uniform segments. Let $\gamma = 2L/K$ and we apply piecewise linear interpolation for each subinterval:

$$e^x \approx ax + b = \frac{e^{x_{i+1}} - e^{x_i}}{x_{i+1} - x_i}(x - x_i) + e^{x_i},$$
$$x_i = -L + \gamma i, \quad i = 0, 1, \ldots, K - 1.$$

The coefficients $a$ and $b$ are precomputed and stored in lookup tables to avoid using the multiplication operator. To minimize memory usage, the coefficient $a$ is quantized to 8-bit fixed-point precision. This design eliminates the need for online exponential computation and enables high-throughput deployment in SNNs.

The PWL-Exp Unit introduces negligible overhead and operates with a cost comparable to a lightweight linear mapping. Its structure is particularly amenable to LUT-based implementations in neuromorphic hardware.

### 4.2. Nonlinear Spiking Function (NLSpiking)

We now build NLSpiking functions. The key observation behind NLSpiking is that a wide class of nonlinear functions used in LLMs shares a common fractional structure: an input-dependent numerator modulated by a positive normalization term. Rather than approximating each nonlinear function independently, we factorize them into a numerator path and a denominator path, each implemented using spike-friendly primitives, and recombine them through the Division Neuron.

This decomposition provides a unified abstraction for seemingly different nonlinearities. Softmax, SiLU, and RMSNorm differ mainly in how the numerator and denominator are constructed, while the overall computational skeleton remains identical.

**Softmax.** Softmax is characterized by a competitive normalization across multiple inputs, where the relative scale between exponentiated activations determines the output distribution. We first stabilize the input using the shift-

invariance property of Softmax:

$$z_i \leftarrow z_i - \max_j z_j + H,$$

ensuring $z_i \in (-\infty, H]$. To suppress negligible contributions, the PWL-Exp Unit outputs zero for inputs below $-H$.

**Numerator:** $\exp(z_i)$ is approximated using the PWL-Exp Unit within the interval $[-H, H]$.

**Denominator:** The summation $\sum_j \exp(z_j)$ is computed through temporal accumulation over $T$ timesteps.

**Output:** The final normalized output is obtained via the Division Neuron.

This formulation avoids runtime exponentiation and division, enabling low-latency spike-based inference with bounded output.

**SiLU.** Unlike Softmax, SiLU is a self-modulated nonlinearity, where the input simultaneously contributes to both the numerator and the normalization term. For SiLU, we constrain the domain to $[-H, H]$ and extend the output linearly beyond this range: $\text{SiLU}(x) \leftarrow x$ for $x > H$, and $\text{SiLU}(x) \leftarrow 0$ for $x < -H$.

**Numerator:** The input $x$ is directly encoded as a spike train.

**Denominator:** The term $1 + \exp(-x)$ is approximated by applying the PWL-Exp Unit to $-x$ and adding 1.

**Output:** The final normalized value is computed via the Division Neuron.

This construction preserves the smooth nonlinearity of SiLU while maintaining spike compatibility.

**RMSNorm.** RMSNorm differs fundamentally from Softmax and SiLU in that its denominator encodes a vector magnitude rather than an activation-dependent gating term. To approximate RMSNorm, we follow the transformed formulation:

**Numerator:** Each $x_i$ is scaled by the constant $\sqrt{d}$ and encoded as a spike train.

**Denominator:** The expression $\sqrt{\sum x_i^2 + \epsilon d}$ is approximated using the PolarNorm Unit.

**Output:** The final normalized value is computed via the Division Neuron.

All nonlinear spiking functions in NLSpiking follow the same modular construction paradigm, relying exclusively on primitive, spike-friendly operations. By explicitly factorizing each nonlinearity into a numerator path and a normalization path, approximation errors are isolated within individual modules rather than entangled across the computation. This modularity enables systematic error analysis

and exposes clear control knobs over accuracy, latency, and resource usage, which is particularly amenable to hardware co-design.

Beyond the specific instances of Softmax, SiLU, and RMSNorm, this construction naturally extends to related normalization-based variants. For example, LayerNorm can be viewed as an RMSNorm augmented with a mean subtraction term, which can be implemented using only additions and accumulations without altering the overall division-based structure. This highlights the generality of NLSpiking as a unifying framework for spike-compatible nonlinear normalization.

A detailed discussion of approximation accuracy, the influence of constants such as $\varepsilon$, and the selection of hyperparameters—including the range bound $H$—is provided in Section 5, where we rigorously derive error bounds and offer principled guidelines for configuring NLSpiking functions.

## 5. Performance Analysis

To validate the effectiveness of the NLSpiking framework, we analyze its performance from two complementary perspectives: *accuracy* and *memory footprint*, aiming to provide a holistic view of how spike-based approximations balance functional fidelity with hardware constraints. Our analysis is guided by a central question: whether the modular decomposition in NLSpiking leads to controlled and predictable approximation errors, or whether errors introduced by individual spike-based operators accumulate uncontrollably. To this end, we derive explicit per-function error bounds that decompose the total approximation error into contributions from exponentiation, division, and norm estimation. Throughout this section we denote by:

$$\varepsilon_{\exp} = \frac{L^2}{2K^2} e^{2L/K}, \quad \Delta = \frac{1}{n}, \quad \varepsilon_{\text{pol}} = \left\lceil \log_2 d \right\rceil 2^{-2n-1},$$

the relative error of PWL-Exp, the quantisation step of the $(T, L)$-Division neuron, and the relative $\ell_2$–norm error of an $n$–step CORDIC tree in the PolarNorm Unit, respectively.

**Theorem 5.1** (Error Bounds for NLSpiking). *For each NL-Spiking function $\tilde{\phi}$, assume the standard spike-based approximations (PWL-Exp, Division Neuron, and PolarNorm) are employed. Then the following per-output error bounds hold:*

(i) **Softmax.** *For every class $i$, $\frac{|\tilde{\phi}_i - \phi_i|}{\phi_i} \leq \frac{2}{1-\varepsilon_{\exp}}\left(\varepsilon_{\exp} + \Delta\right)$.*

(ii) **SiLU.** *For every $x \in [-H, H]$, $|\tilde{\phi}(x) - \phi(x)| \leq |x| \frac{2\varepsilon_{\exp}}{1-\varepsilon_{\exp}} + |x|\Delta$.*

(iii) **RMSNorm.** *For each coordinate $i$, $\frac{|\tilde{\phi}_i - \phi_i|}{\phi_i} \leq \frac{\varepsilon_{\text{pol}} + \Delta}{1-\varepsilon_{\text{pol}}} + \sqrt{d}\,\Delta$.*

A key observation from Theorem 5.1 is that, for all three nonlinearities, the total approximation error decomposes additively into a small number of well-isolated terms. These bounds demonstrate that NLSpiking functions achieve high-precision approximations across core nonlinearities. Each error term remains tightly controlled, stemming from well-isolated sources: lookup-based exponentiation, spike-based division, and recursive norm estimation. We empirically validate these guarantees in the next section. The proof is available in the Appendix A.

**Memory Efficiency.** It is worth noting that Theorem 5.1 directly exposes the trade-off between approximation accuracy and memory footprint. In particular, the PWL-Exp error $\varepsilon_{\exp}$ depends only on the number of segments $K$, which determines the size of the lookup table. Compared to traditional floating-point computation, NLSpiking only requires storing a compact set of lookup entries: $K$ values of 8-bit and 16-bit precision in total. This is significantly smaller than typical table-based methods, which often rely on large floating-point tables. Such reduction is critical for deployment on spiking neuromorphic chips, which usually operate under strict on-chip memory constraints.

**Remark 5.2** (Recommended Setting). The error bounds reveal a clear accuracy–memory trade-off governed primarily by $\varepsilon_{\exp}$. In practice, requiring $\varepsilon_{\exp} < 10^{-2}$ is sufficient to make the approximation error negligible compared to quantization noise in low-precision inference. We therefore recommend $H = 5$ and $K = 64$, which yields $\varepsilon_{\exp} \leq 3.63 \times 10^{-3}$ while keeping the lookup table compact.

## 6. Experiments

In this section, we present a comprehensive evaluation of the proposed method from both the operator-level and the model-level perspectives, together with a sensitivity analysis of key hyperparameters. More detailed experiments are deferred to the Appendix B. Furthermore, while nonlinear operators generally incur lower computational cost than linear layers in practice, we also report the counts of MAC, AC, and shift operations in Appendix B.4 to provide. a reference for energy analysis.

### 6.1. Function-Level Evaluation.

**Baseline.** Our evaluation is conducted strictly at the operator level. For each target function, we compare NLS-based operators against representative approximation implementations commonly adopted for the same operator in quantized or efficient inference settings. Most of these operator instantiations are adapted from prior work on surrogate/nonlinear approximations and extreme quantization, including Sorbet (Tang et al., 2025), XNOR-Net (Rastegari et al., 2016), and DoReFa-Net (Zhou et al., 2016), and we additionally in-

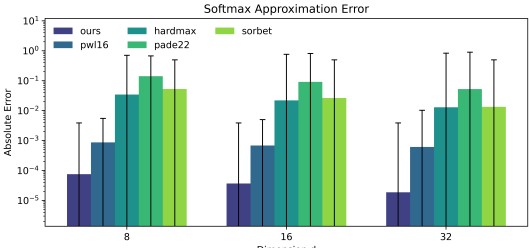
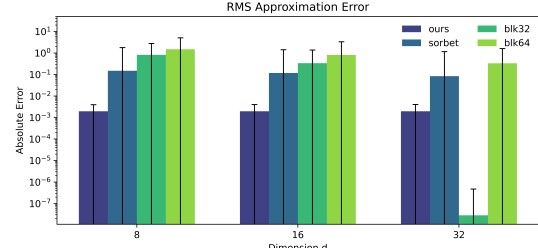

*(a)* Operator-level errors for Softmax approximations under 8-bit quantization.

*(b)* Operator-level errors for RMSNorm approximations under 8-bit quantization.

*Figure 3.* Operator-level errors under 8-bit quantization. Error bars indicate the gap between mean and maximum absolute error. NLS-Softmax achieves the lowest mean error across dimensions while keeping bounded maximum error under integer-only implementation, and NLS-RMS yields lower mean errors than blockwise and Sorbet baselines with stable performance across dimensions.

clude classical numerical approximations for completeness.

For **SiLU**, we include XNOR-style binary thresholding, DoReFa-style low-bit uniform quantization (DoReFa-4b), surrogate nonlinearities (ReLU and hard-swish), and piecewise-linear (PWL) sigmoid-based approximations with 16 and 64 segments. These baselines span the spectrum from extreme low-cost discretization to higher-precision numerical approximations.

For **Softmax**, we compare against hardmax, surrogate-based approximations (Sorbet), a Padé [2/2] rational approximation, and a 16-segment PWL exponential baseline, which approximate the exponential and normalization steps in efficient attention.

For **RMSNorm**, we include Sorbet-style surrogates and blockwise RMS approximations with block sizes of 32and 64, which are commonly used to reduce reduction cost in practice. All methods are treated as isolated operator approximations and evaluated.

**Operator-level Approximation Results.** Figures 3a, 3b, 4a summarize operator-level approximation errors under 8-bit quantization.

For **SiLU** (Fig. 4a, $x \in [-5, 5]$), NLS-SiLU achieves the lowest mean error among training-free baselines, while keeping maximum error comparable to a strong 64-segment PWL-sigmoid; Sorbet and hard-swish deviate more, and DoReFa-4b/XNOR exhibit both high mean and peak errors.

For **Softmax** (Fig. 3a, varying input dimension $d$), NLS-Softmax consistently yields the lowest mean error across all $d$, with maximum error effectively bounded by the 8-bit grid, whereas Padé/PWL may approach similar mean accuracy but suffer larger maximum error and variance at higher dimensions, and Sorbet/hardmax remain substantially worse.

For **RMSNorm** (Fig. 3b), blockwise RMS (block size 32/64) can be accurate only when $d$ aligns with the block partition, but becomes unstable otherwise; in contrast, NLS-RMS maintains consistently low mean error across all tested dimensions.

Overall, NLS-based operators provide stable, quantization-robust approximations, while enabling an efficient deployment that shares a single lookup table across Softmax, SiLU, and RMSNorm.

### 6.2. Ablation and Function Sensitivity Analysis

Figure 4b shows the sensitivity of NLSpike-SiLU when $H$ is varied from 3 to 10. We observe that both mean and maximum approximation errors remain extremely small within moderate ranges (e.g., $H = 3, 4, 5$). However, when $H$ grows larger, the maximum error increases rapidly, reflecting the growing difficulty of approximating the extreme tails of the activation. This suggests that unnecessarily large intervals harm robustness without improving the error in the practically relevant region. In contrast, Figure 4c reports the sensitivity of NLSpike-Softmax (fixed $d = 64$). Here, enlarging $H$ consistently reduces both mean and maximum errors, because clipping less aggressively preserves the exponential scaling inside the softmax. Nevertheless, small $H$ values ($\leq 4$) already induce non-negligible errors that may accumulate at the layer level.

### 6.3. Model-Level Evaluation.

We evaluate on two categories of large language models. The first category comprises SNN models produced by ANN-to-SNN conversion pipelines, including:

**SpikeLLM:** (Xing et al., 2025) the first spiking large language model using ANN-to-SNN.

**SpikeZIP:** (You et al., 2024) a novel ANN-to-SNN conversion method used in BERT and ViT.

We use their conversion pipelines as released and perform operator replacement *only after* the conversion stage,

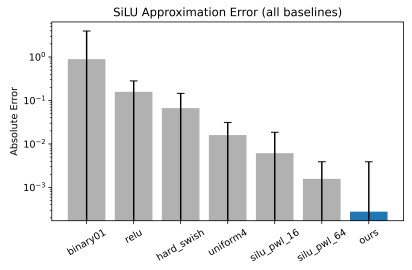

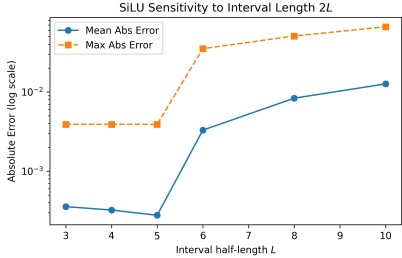

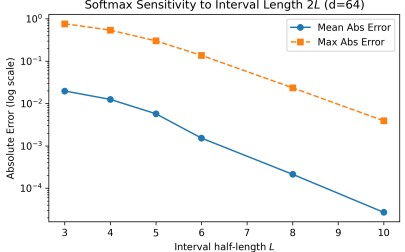

*(a)* SiLU approximation error (8-bit).  *(b)* NLS-SiLU sensitivity to $2L$.  *(c)* NLS-Softmax sensitivity ($d = 64$).

*Figure 4.* **Left**: SiLU approximation errors across baselines. **Middle–Right**: sensitivity of NLS-SiLU and NLS-Softmax to the clipping interval length $2L$. Excessively large intervals increase SiLU maximum error, while overly small intervals lead to significant Softmax deviations. We recommend $L = 5$ as the default setting.

*Table 1.* Model-level accuracy before and after operator replacement. $\Delta$ denotes the change from the original operator. NLSpike results are highlighted for clarity.

| Source | Model | Operator | WinoGrande | HellaSwag | ArcC | ArcE | PIQA | Avg. Acc. |
|---|---|---|---|---|---|---|---|---|
| ANN | LLaMA-3-8B (Dubey et al., 2024) | Original | 0.736 | 0.792 | 0.542 | 0.776 | 0.807 | 0.730 |
| | | NLSpike ($\Delta$) | **-0.008** | **-0.000** | **+0.001** | **+0.000** | **-0.004** | **-0.003** |
| | LLaMA-2-7B (Touvron et al., 2023) | Original | 0.693 | 0.762 | 0.451 | 0.737 | 0.788 | 0.686 |
| | | NLSpike ($\Delta$) | **-0.006** | **-0.001** | **+0.001** | **-0.003** | **+0.001** | **-0.002** |
| | Mistral-7B (Jiang et al., 2023) | Original | 0.736 | 0.844 | 0.544 | 0.717 | 0.779 | 0.724 |
| | | NLSpike ($\Delta$) | **+0.001** | **-0.001** | **-0.003** | **+0.000** | **+0.004** | **+0.000** |
| | Qwen3-8B (Yang et al., 2025) | Original | 0.720 | 0.786 | 0.565 | 0.800 | 0.797 | 0.734 |
| | | NLSpike ($\Delta$) | **-0.005** | **+0.002** | **+0.039** | **+0.033** | **-0.001** | **+0.014** |
| SpikeLLM$_{T=2,W2A16}$ | LLaMA-2-7B (Touvron et al., 2023) | Original | 0.528 | 0.499 | 0.284 | 0.419 | 0.657 | 0.477 |
| | | NLSpike ($\Delta$) | **-0.004** | **-0.002** | **+0.002** | **-0.001** | **+0.003** | **-0.000** |
| | LLaMA-2-13B (Touvron et al., 2023) | Original | 0.572 | 0.580 | 0.304 | 0.445 | 0.677 | 0.516 |
| | | NLSpike ($\Delta$) | **-0.003** | **-0.001** | **+0.001** | **+0.002** | **-0.002** | **-0.001** |

*Table 2.* Performance of a BERT model converted via the SpikeZIP ANN2SNN pipeline ($T = 64$) before and after applying NLSpike.

| Operator type | MR | SST-2 | Subj | SST-5 | Avg. Acc. |
|---|---|---|---|---|---|
| Original | 0.881 | 0.904 | 0.943 | 0.500 | 0.807 |
| **NLSpike($\Delta$)** | **-0.001** | **+0.006** | **-0.003** | **+0.009** | **+0.003** |

without modifying any pipeline components. We use a LayerNorm-based NLSpike variant for the BERT model in our experiments, with activations implemented via a piecewise linear approximation.

The second category comprises standard ANN-based LLMs that are not explicitly covered by existing ANN-to-SNN conversion pipelines, including LLaMA-3-8B, LLaMA-2-7B, Mistral-7B, and Qwen3-8B. We collapse the temporal dimension of the division neuron into a single-step representation and interface it directly with the ANN computation graph, while keeping the architecture, all linear layers, pretrained parameters, and inference hyperparameters unchanged, and without any retraining. This controlled protocol isolates the impact of our operator implementations at the model level and highlights their potential as building blocks for future SNN-based LLM designs.

As shown in Tabel 1, 2, model-level evaluation results show that NLSpike first validates its core design objective in ANN-to-SNN converted models. For SNNs produced by the SpikeLLM and SpikeZIP pipelines, we conduct evaluation without modifying any conversion procedures and replace the nonlinear operators only after the conversion stage. The results indicate that performance changes across all tasks are negligible, with average accuracy remaining stable, demonstrating that NLSpike can be directly loaded as an independent operator into existing ANN2SNN pipelines while maintaining compatibility with established temporal dynamics and spike-based inference. Building on this, we further evaluate NLSpike on standard LLMs without conversion, where it similarly introduces no noticeable performance degradation and even yields positive gains on certain reasoning tasks, suggesting both strong robustness.

### 6.4. Latency Analysis

We provide a hardware-aware latency analysis under practical neuromorphic execution models. Digital neuromorphic hardware typically consists of neuromorphic cores and embedded processors (Davies et al., 2018; Ma et al., 2024). Neuromorphic cores mainly support lightweight arithmetic

*Table 3.* Latency comparison under practical neuromorphic execution models.

| Method | Nonlinear / Special-Function Calls | Data Movement | Time-Step Latency |
|---|---|---|---|
| SpikeZIP (Xing et al., 2025) | $T\times$ nonlinear evaluations | $\mathcal{O}(T)$ cross-domain | **0** |
| SpikeLLM (You et al., 2024) | $1\times$ nonlinear evaluation | $\mathcal{O}(1)$ cross-domain | $T$ |
| NLSpike (Ours) | $n\times$ **shift-add / LUT calls** | **0 (in-core)** | $T$ |

datapaths such as LUT and shift-add operations, and are generally not optimized for floating-point nonlinear computation (Ma et al., 2024). Consequently, nonlinear operators are often executed on external processors, introducing substantial cross-domain data movement overhead. We analyze latency from the perspective of temporal execution structures in ANN-to-SNN nonlinear operator pipelines.

**SpikeLLM-style pipelines** (Xing et al., 2025). Nonlinear operators are evaluated after spike accumulation over a temporal window, followed by spike emission. This introduces a two-window execution structure for each operator. Our method follows the same temporal structure and therefore does not introduce additional time-step latency compared to this class of methods.

**SpikeZIP-style pipelines** (You et al., 2024). Nonlinear operators are evaluated incrementally at every time step: $O(t) = o(t) - o(t - 1)$. This requires repeated nonlinear evaluations across $T$ time steps, resulting in $\mathcal{O}(T)$ nonlinear computations.

As discussed in (Li et al., 2020), practical neuromorphic deployment cost is often dominated by cross-domain data movement rather than arithmetic computation itself. In Tabel 3, SpikeZIP-style pipelines achieve zero time-step latency, but require repeated nonlinear evaluations together with $\mathcal{O}(T)$ cross-domain communication overhead. SpikeLLM reduces nonlinear evaluations to one, but still relies on cross-domain nonlinear execution. In contrast, our method executes nonlinear operators entirely within neuromorphic cores using shift-add and LUT operations, thereby eliminating cross-domain data movement while maintaining the same temporal execution structure as SpikeLLM.

## 7. Conclusion and Limitations

This paper presents a plug-and-play framework that approximates key Transformer nonlinearities with LIF neuron populations, using only shift-and-scale operations and largely avoiding floating-point computation, making it compatible with existing ANN-to-SNN pipelines. A current limitation is the lack of end-to-end LLM deployment on real spiking hardware due to operator, memory, and accuracy–latency constraints. Future work will focus on end-to-end deployment on real spiking hardware.

## Impact Statement

This paper presents work whose goal is to advance the field of Machine Learning. There are many potential societal consequences of our work, none which we feel must be specifically highlighted here.

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

# A. Proof of Theorem1

Based on the method introduced in the previous section, we have derived a set of spike-compatible NLS-functions intended for application in the forward propagation of the spike-LLM. We now estimate the approximation errors of the three NLS-functions used in this work: NLS-Softmax, NLS-SiLU, and NLS-RMSNorm. First, we need some necessary lemmas.

**Lemma A.1** (PWL-Exp Unit Relative Error Bound). *Let $\tilde{e}^x$ be the piecewise-linear approximation to $e^x$ on $[-H, H]$ obtained by dividing the interval into $K$ equal subintervals of width*

$$h = \frac{2H}{K},$$

*and interpolating linearly on each. Then for every $x \in [-H, H]$,*

$$\left| \frac{\tilde{e}^x - e^x}{e^x} \right| \leq \frac{h^2}{8} e^h = \frac{\left(\frac{2H}{K}\right)^2}{8} e^{2H/K}.$$

*Especially, when $H = 5, K = 64$, we have:*

$$\left| \frac{\tilde{e}^x - e^x}{e^x} \right| \leq \frac{h^2}{8} e^h = \frac{\left(\frac{5}{32}\right)^2}{8} e^{5/32} \approx 3.63 \times 10^{-3}.$$

**Lemma A.2.** *(Volder, 1959)[Pairwise CORDIC Relative Error Bound] Let $a, b \in \mathbb{R}$ and $r = \sqrt{a^2 + b^2}$. Perform $n$ steps of CORDIC in vectoring mode with angles $\phi_i = \arctan(2^{-i})$ for $i = 0, 1, \ldots, n - 1$, and apply the exact scale correction*

$$K_n^{-1} = \prod_{i=0}^{n-1} \frac{1}{\sqrt{1 + 2^{-2i}}}.$$

*If $\tilde{r}$ denotes the CORDIC output after correction, then*

$$\frac{|\tilde{r} - r|}{r} \leq 2^{-2n-1}.$$

**Lemma A.3** (Binary-Tree CORDIC Accumulation Error). *To compute*

$$R = \sqrt{\sum_{j=1}^{D} v_j^2 + \varepsilon D},$$

*group the $D$ values into $\lceil D/2 \rceil$ pairs, apply Lemma A.2 to each pair (using the same $n$-step CORDIC), and then recursively merge the resulting radii in a balanced binary tree of height $\ell = \lceil \log_2 D \rceil$. Denote the final CORDIC result by $\tilde{R}$. Then*

$$\frac{|\tilde{R} - R|}{R} \leq \ell \cdot 2^{-2n-1}.$$

After discussing the error bounds of PWL-Exp Unit and PolarNorm Unit, now we can focus on the errors of NLS-Softmax, NLS-SiLU, and NLS-RMSNorm.

*Proof of Lemma A.3.* Let the pairwise CORDIC relative error bound be $\delta := 2^{-2n-1}$. We analyze the error propagation layer by layer through the CORDIC binary tree.

**Base layer (Layer 1):** Group the $D$ inputs into $\lceil D/2 \rceil$ pairs. Apply Lemma A.2 to each pair, yielding approximate radii $\tilde{r}_k^{(1)}$ with relative error $\leq \delta$.

**Inductive step (Layer $t$):** Assume inputs to layer $t$ have maximum relative error $\leq (t - 1)\delta$. Merging two such values gives:

$$\tilde{r}^{(t)} = \sqrt{(\tilde{r}_1^{(t-1)})^2 + (\tilde{r}_2^{(t-1)})^2} \cdot (1 + e_{\text{CORDIC}}),$$

where $|e_{\text{CORDIC}}| \leq \delta$ (by Lemma A.2).

Each $\tilde{r}^{(t-1)}$ differs from its true value $r^{(t-1)}$ by at most $(t-1)\delta$, so the total error in computing $\tilde{r}^{(t)}$ is at most:

$$|e^{(t)}| \leq (t-1)\delta + \delta = t\delta.$$

**Final result.** The CORDIC reduction forms a complete binary tree whose leaves are the $D$ original vectors. At every layer the number of active nodes is at most halved (merging each pair into one). After $t$ layers the node count is therefore at most $D/2^{t}$. We need enough layers so that only one node remains:

$$\frac{D}{2^{\ell}} \leq 1 \quad \Longrightarrow \quad \ell \geq \log_2 D.$$

Taking the smallest integer that meets the inequality yields

$$\ell = \lceil \log_2 D \rceil.$$

Consequently, after $\ell$ layers the relative error of the final radius satisfies

$$\frac{|\tilde{R} - R|}{R} \leq \ell\delta = \ell 2^{-2n-1}.$$

$\square$

**Theorem A.4** (Error Bound of NLS-Softmax). *Let $\mathbf{z} = (z_1, \ldots, z_d)$ be the shifted and clipped logits*

$$-H \leq z_i \leq L, \qquad i = 1, \ldots, K,$$

*and write*

$$p_i = \frac{e^{z_i}}{\sum_{j=1}^{d} e^{z_j}} \quad \text{for the exact softmax, and} \quad \tilde{p}_i = \frac{\tilde{e}^{z_i}}{\sum_{j=1}^{d} \tilde{e}^{z_j}} + \delta_i$$

*for the NLS-Softmax output, where*

\* $\tilde{e}^{\cdot}$ *is the PWL-Exp approximation of Lemma 1 whose relative error satisfies $|\tilde{e}^x - e^x| \leq \varepsilon_{\exp} e^x$ with $\varepsilon_{\exp} = \frac{\left(\frac{2K}{K}\right)^2}{8} e^{2K/K}$;*

\* *the final reciprocal is produced by a $(T, L)$-Division neuron whose quantisation step is $\Delta = \frac{1}{TL}$, so that $|\delta_i| \leq \Delta p_i$ for every $i$.*

*Then, for every class $i$,*

$$\boxed{\frac{|\tilde{p}_i - p_i|}{p_i} \leq \frac{2}{1 - \varepsilon_{\exp}}(\varepsilon_{\exp} + \Delta).}$$

*Proof.* **Step 1: bound the exponential approximations.** By Lemma 1,

$$(1 - \varepsilon_{\exp})e^{z_i} \leq \tilde{e}^{z_i} \leq (1 + \varepsilon_{\exp})e^{z_i}, \qquad i = 1, \ldots, d. \tag{13}$$

**Step 2: bound the denominator.** Summing (13) yields

$$(1 - \varepsilon_{\exp})S \leq \tilde{S} := \sum_j \tilde{e}^{z_j} \leq (1 + \varepsilon_{\exp})S, \qquad S = \sum_j e^{z_j}.$$

**Step 3: ratio perturbation.** Define the "ideal" NLS-Softmax without quantisation, $\hat{p}_i = \tilde{e}^{z_i}/\tilde{S}$. Using the standard perturbation inequality[1] and (13),

$$\frac{|\hat{p}_i - p_i|}{p_i} \leq \frac{2\varepsilon_{\exp}}{1 - \varepsilon_{\exp}}.$$

---

[1] For positive $a, b$ and errors $|\delta a| \leq \varepsilon a$, $|\delta b| \leq \varepsilon b$, one has $\left|(a + \delta a)/(b + \delta b) - a/b\right| \leq 2\varepsilon\, a/b\,(1 - \varepsilon)^{-1}$.

**Step 4: add the division-neuron quantisation.** The Division neuron introduces an extra additive error $|\delta_i| \leq \Delta$, so

$$|\tilde{p}_i - p_i| \leq |\tilde{p}_i - \hat{p}_i| + |\hat{p}_i - p_i| \leq \left(\Delta + \frac{2\varepsilon_{\exp}}{1 - \varepsilon_{\exp}}\right) p_i.$$

Dividing both sides by $p_i$ proves the claimed bound. $\qquad\square$

**Theorem A.5** (Error Bound of NLS-SiLU). *Assume the input is clipped to $x \in [-5, 5]$ before NLS-SiLU is applied. Let*

$$f(x) = \text{SiLU}(x) = \frac{x}{1 + e^{-x}}, \qquad \tilde{f}(x) = x\,\tilde{\sigma}(x) + \delta_{mul},$$

*where*

\* $\tilde{\sigma}(x) = \dfrac{1}{1 + \tilde{e}^{-x}} + \delta_{div}$ *is obtained from the PWL-Exp approximation $\tilde{e}^{\cdot}$ of Lemma 1 (relative error $\varepsilon_{\exp} = 3.63 \times 10^{-3}$) followed by a $(T, L)$-Division neuron whose quantisation step is $\Delta = 1/(TL)$;*

\* $\delta_{div}$ *and $\delta_{mul}$ are, respectively, the additive errors introduced by the Division neuron and by the spike–time multiplication, both satisfying $|\delta_{div}|, |\delta_{mul}| \leq \Delta$.*

*Then, for every $x \in [-5, 5]$,*

$$\boxed{\left|\tilde{f}(x) - f(x)\right| \leq |x|\,\frac{2\varepsilon_{\exp}}{1 - \varepsilon_{\exp}} + |x|\,\Delta}$$

*and in particular, with $(T, L) = (16, 256)$ so that $\Delta = 2^{-12} \approx 2.44 \times 10^{-4}$,*

$$\left|\tilde{f}(x) - f(x)\right| \leq 0.038 \qquad \text{for all } x \in [-5, 5].$$

*Proof.* **1. Bounding the sigmoid approximation.** Because $\sigma(x) = 1/(1 + e^{-x})$ is Lipschitz-continuous on $[-5, 5]$ with Lipschitz constant at most $1/4$, the PWL-Exp error of Lemma 1 implies

$$\left|\hat{\sigma}(x) - \sigma(x)\right| = \left|\frac{1}{1 + \tilde{e}^{-x}} - \frac{1}{1 + e^{-x}}\right| \leq \frac{2\varepsilon_{\exp}}{1 - \varepsilon_{\exp}}\,\sigma(x),$$

where $\hat{\sigma}(x) = 1/(1 + \tilde{e}^{-x})$ is the *ideal* reciprocal without quantisation.

**2. Adding division-neuron quantisation.** The Division neuron produces $\tilde{\sigma}(x) = \hat{\sigma}(x) + \delta_{div}$ with $|\delta_{div}| \leq \Delta$, hence

$$\left|\tilde{\sigma}(x) - \sigma(x)\right| \leq \frac{2\varepsilon_{\exp}}{1 - \varepsilon_{\exp}}\,\sigma(x) + \Delta. \qquad (14)$$

Therefore, using (14),

$$\left|\tilde{f}(x) - f(x)\right| \leq |x|\,\frac{2\varepsilon_{\exp}}{1 - \varepsilon_{\exp}} + |x|\,\Delta,$$

which is the claimed bound. For $(T, L) = (256, 16)$ we substitute $\varepsilon_{\exp} = 3.63 \times 10^{-3}$ and $\Delta \approx 2.44 \times 10^{-4}$ to obtain the numerical value. $\qquad\square$

**Theorem A.6** (Error Bound of NLS-RMSNorm). *Let $\mathbf{x} = (x_1, \ldots, x_d) \in \mathbb{R}^d$ and define*

$$R = \sqrt{\frac{1}{d}\sum_{j=1}^{d} x_j^2 + \varepsilon}, \qquad y_i = \frac{x_i}{R}, \quad i = 1, \ldots, d.$$

*Denote by $\tilde{R}$ the output of the **PolarNorm** unit that uses an $n$–step CORDIC in a balanced binary tree of height $\ell = \lceil \log_2 d \rceil$ and let*

$$\varepsilon_{pol} := \ell\,2^{-2n-1} \quad \text{so that} \quad \left|\frac{\tilde{R} - R}{R}\right| \leq \varepsilon_{pol} \quad \text{(Lemma A.3)}.$$

*Next, let the $(T, L)$–Division neuron produce*

$$\tilde{q} = \frac{1}{\tilde{R}} + \delta_{div}$$

*with quantisation step*

$$\Delta := \frac{1}{TL} \quad (or \ |\delta_{div}|_{\max}),$$

*and finally obtain the NLS-RMSNorm output*

$$\tilde{y}_i = x_i \tilde{q} + \delta_{mul}, \qquad |\delta_{mul}| \le \Delta.$$

*Then, for each coordinate $i = 1, \ldots, d$, we have*

$$\boxed{\frac{|\tilde{y}_i - y_i|}{|y_i|} \ \le \ \frac{\varepsilon_{pol} + \Delta}{1 - \varepsilon_{pol}} \ + \ \frac{\Delta}{|x_i|}R}$$

*and, whenever $|x_i| \ge \sqrt{\varepsilon}$ (the usual case in practice), the last term satisfies*

$$\frac{\Delta}{|x_i|}R \le \sqrt{d} \cdot \Delta.$$

*Proof.* We decompose the total error as

$$\tilde{y}_i - y_i = x_i\big(\tilde{q} - \frac{1}{R}\big) = x_i\Big(\frac{1}{\tilde{R}} - \frac{1}{R}\Big) + x_i\delta_{\mathrm{div}}.$$

**(i) Reciprocal of $\tilde{R}$:** Using the approximation

$$\tilde{R} = R(1 + \eta), \quad |\eta| \le \varepsilon_{\mathrm{pol}},$$

we have the following identity:

$$\left|\frac{1}{\tilde{R}} - \frac{1}{R}\right| = \frac{|\eta|}{1 + \eta} \cdot \frac{1}{R} \le \frac{\varepsilon_{\mathrm{pol}}}{1 - \varepsilon_{\mathrm{pol}}} \cdot \frac{1}{R}.$$

This result follows from the first-order approximation and ensures that the error is controlled by the CORDIC approximation precision.

**(ii) Division-neuron quantisation:** Since the quantisation error is bounded by

$$|x_i\delta_{\mathrm{div}}| \le |x_i|\Delta,$$

we can combine these terms to get:

$$|\tilde{y}_i - y_i| \le |x_i| \cdot \frac{\varepsilon_{\mathrm{pol}}}{1 - \varepsilon_{\mathrm{pol}}} \cdot \frac{1}{R} + |x_i|\Delta.$$

Dividing both sides by $|y_i| = \frac{|x_i|}{R}$ gives:

$$\frac{|\tilde{y}_i - y_i|}{|y_i|} \le \frac{\varepsilon_{\mathrm{pol}}}{1 - \varepsilon_{\mathrm{pol}}} + \frac{\Delta}{|x_i|}R.$$

Thus, we obtain the final error bound:

$$\frac{|\tilde{y}_i - y_i|}{|y_i|} \le \frac{\varepsilon_{\mathrm{pol}} + \Delta}{1 - \varepsilon_{\mathrm{pol}}} + \frac{\Delta}{|x_i|}R.$$

Finally, when $|x_i|$ is sufficiently large (i.e., $|x_i| \ge \sqrt{\varepsilon}$), we can bound the term:

$$\frac{\Delta}{|x_i|}R \le \sqrt{d} \cdot \Delta,$$

because $R$ is bounded by a factor of $\sqrt{d}$, as derived from the sum over all $x_j$. $\qquad \square$

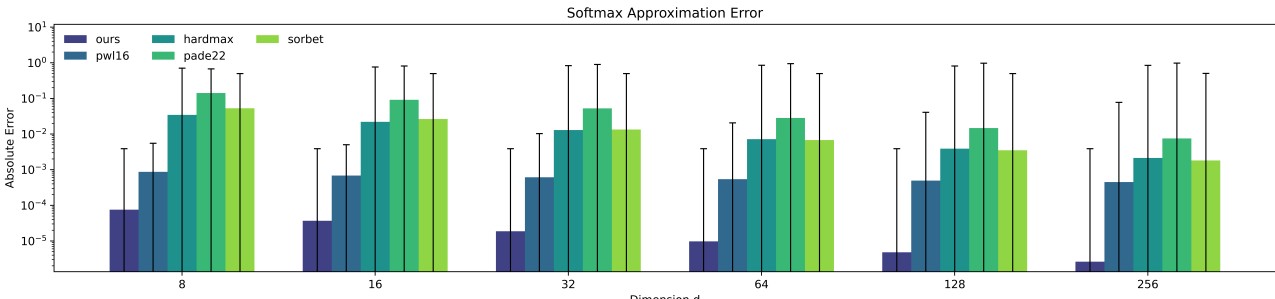

*(a)* Operator-level errors for Softmax approximations under 8-bit quantization.

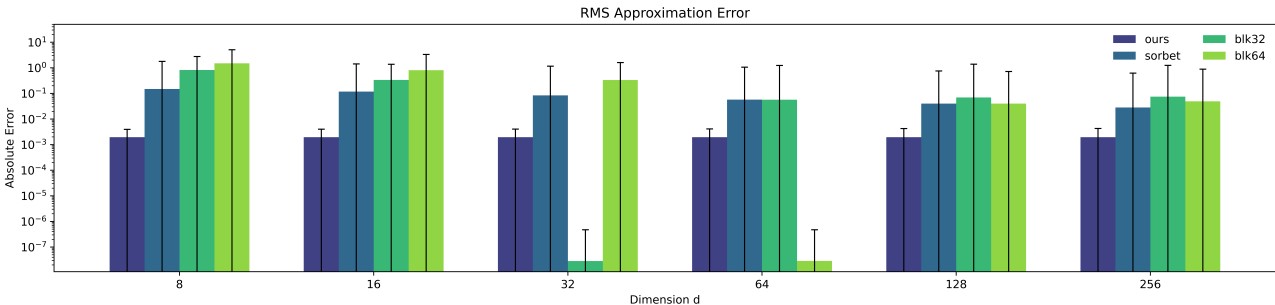

*(b)* Operator-level errors for RMSNorm approximations under 8-bit quantization.

*Figure 5.* Operator-level errors under 8-bit quantization. Error bars indicate the gap between mean and maximum absolute error. ES-Softmax achieves the lowest mean error across dimensions while keeping bounded maximum error under integer-only implementation, and ES-RMS yields lower mean errors than blockwise and Sorbet baselines with stable performance across dimensions.

# B. More Experiments

## B.1. Function-Level Evaluation

In addition to the errors provided in the main text, we present here the errors for d=8, 16, 32, 64, 128, and 256 in Figure 5. The results demonstrate that NLSpike consistently maintains its advantage.

## B.2. Module-Level Evaluation.

We take LLaMA3-QCFS (with 8-bit weights and activations, 8B scale) as the base model and evaluate three different conversion strategies: (1) a direct ANN-to-SNN conversion without approximation, (2) conversion using the NLS framework in time-independent form (NLS), and (3) conversion using the time-dependent form (NLS-TDF). For evaluation, we randomly sample 1,000 tasks from the MMLU dataset and run inference using each of the three converted models to assess their module-level performance under different design configurations.

Figure 6 presents the mean $L_2$ error on 1,000 MMLU samples across different time steps $T \in \{2, 4, 8\}$. The results show that both NLS and NLS-TDF maintain approximation errors comparable to the direct ANN-to-SNN conversion baseline. Notably, as $T$ increases, the gap among all three configurations narrows, suggesting that spike accumulation compensates for early approximation gaps. These results validate the functional fidelity of our NLS-based modules under typical SNN execution settings.

## B.3. Model-Level Evaluation.

To assess the scalability and effectiveness of our framework at the full model level, we conduct evaluations on LLaMA2-7B, LLaMA3-8B, and LLaMA3-70B. Each model is first converted into spiking form via ANN-to-SNN conversion under two quantization settings: W6A6 and W8A8. For each setting, we compare two conversion variants: a baseline SNN without approximation and NN with NLSpike. All evaluations are performed at time steps $T \in \{1, 2, 4\}$ to capture both low-latency and high-accuracy behaviors.

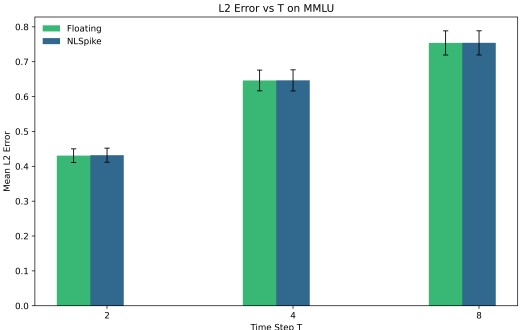

*Figure 6.* Module-Level approximation error in MMLU datasets.

*Table 4.* Performance on LLaMA Models. We report *acc* for WinoGrande and *acc_norm* for HellaSwag, ArcE, and PIQA.

| Model | Method | T | Precision | WinoGrande | HellaSwag | ArcC | ArcE | PIQA | Avg. Acc. |
|---|---|---|---|---|---|---|---|---|---|
| | PrefixQ | - | W6A6/W8A8 | 70.09 / 70.24 | 74.06 / 76.44 | 44.80 / 45.99 | 73.11 / 73.02 | 77.15 / 78.35 | 67.84 / 68.81 |
| | DuQ | - | W6A6/W8A8 | 67.88 / 66.69 | 72.64 / 72.81 | 40.53 / 40.36 | 53.07 / 53.37 | 77.15 / 77.20 | 62.25 / 62.09 |
| | SNN | 1 | W6A6/W8A8 | 70.09 / 70.24 | 74.06 / 76.44 | 44.80 / 45.99 | 73.11 / 73.02 | 77.15 / 78.35 | 67.84 / 68.81 |
| | **NLS** | 1 | W6A6/W8A8 | 67.48 / 68.35 | 73.87 / 76.23 | 44.71 / 46.50 | 73.32 / 73.86 | 76.44 / 78.62 | 67.16 / 68.71 |
| Llama 2 | SNN | 2 | W6A6/W8A8 | 69.06 / 69.93 | 74.23 / 76.45 | 44.88 / 45.90 | 72.98 / 72.90 | 75.68 / 78.45 | 67.37 / 68.73 |
| 7B | **NLS** | 2 | W6A6/W8A8 | 69.53 / 69.61 | 74.17 / 76.46 | 44.45 / 46.33 | 73.06 / 73.11 | 76.77 / 78.35 | 67.60 / 68.77 |
| | SNN | 4 | W6A6/W8A8 | 69.46 / 69.85 | 74.11 / 76.59 | 45.14 / 46.16 | 72.98 / 73.40 | 76.82 / 78.24 | 67.70 / 68.85 |
| | **NLS** | 4 | W6A6/W8A8 | 68.27 / 70.01 | 73.39 / 76.34 | 43.34 / 46.33 | 72.77 / 73.95 | 76.66 / 78.51 | 66.89 / 69.03 |
| | PrefixQ | - | W6A6/W8A8 | 71.82 / 73.09 | 77.61 / 78.96 | 50.94 / 53.75 | 75.59 / 77.99 | 77.69 / 80.47 | 70.73 / 72.85 |
| | DuQ | - | W6A6/W8A8 | 67.88 / 73.56 | 72.64 / 79.07 | 40.53 / 53.24 | 53.07 / 77.95 | 77.15 / 80.25 | 62.25 / 72.81 |
| | SNN | 1 | W6A6/W8A8 | 71.82 / 73.09 | 77.61 / 78.96 | 50.94 / 53.75 | 75.59 / 77.99 | 77.69 / 80.47 | 70.73 / 72.85 |
| | **NLS** | 1 | W6A6/W8A8 | 74.11 / 73.72 | 77.40 / 79.04 | 49.40 / 53.41 | 75.59 / 77.65 | 77.75 / 80.36 | 70.85 / 72.84 |
| Llama 3 | SNN | 2 | W6A6/W8A8 | 71.82 / 73.16 | 77.75 / 79.01 | 47.27 / 53.75 | 75.21 / 77.86 | 75.84 / 79.98 | 69.58 / 72.75 |
| 8B | **NLS** | 2 | W6A6/W8A8 | 72.22 / 73.24 | 77.58 / 78.83 | 48.89 / 53.50 | 75.63 / 77.57 | 77.86 / 80.03 | 70.44 / 72.63 |
| | SNN | 4 | W6A6/W8A8 | 70.40 / 73.32 | 77.65 / 78.91 | 48.98 / 53.58 | 74.33 / 80.43 | 75.90 / 79.98 | 69.45 / 73.24 |
| | **NLS** | 4 | W6A6/W8A8 | 71.19 / 73.09 | 77.34 / 78.81 | 48.81 / 53.75 | 74.37 / 77.90 | 76.77 / 80.30 | 69.69 / 72.77 |
| | PrefixQ | - | W8A8 | 79.32 | 85.65 | 62.37 | 82.79 | 84.11 | 78.85 |
| | DuQ | - | W8A8 | 80.82 | 84.83 | 63.48 | 85.73 | 84.39 | 79.85 |
| Llama 3 | SNN | 1 | W8A8 | 79.32 | 85.65 | 62.37 | 82.79 | 84.11 | 78.85 |
| 70B | **NLS** | 1 | W8A8 | 78.85 | 85.71 | 62.54 | 82.20 | 83.90 | 78.64 |
| | SNN | 2 | W8A8 | 79.48 | 85.70 | 62.88 | 82.87 | 83.90 | 78.97 |
| | **NLS** | 2 | W8A8 | 79.08 | 85.60 | 62.88 | 82.62 | 83.90 | 78.82 |

Table 4 summarizes the end-to-end performance of our ES-converted models on five representative language understanding tasks: Winogrande, HellaSwag, ARC-Challenge, ARC-Easy, and PIQA. We evaluate both standard SNN baselines and ES-based conversions on LLaMA2-7B and LLaMA3-8B under W6A6 and W8A8 quantization settings, across time steps $T \in \{1, 2, 4\}$, and on LLaMA3-70B under W8A8 across time steps $T \in \{1, 2\}$.

As shown, NLSpike models achieve accuracy comparable to direct SNN conversions across most benchmarks. At low time steps ($T = 1$), NLSpike retains competitive accuracy while remaining fully compatible with integer-only execution. As $T$ increases, the performance gap among methods narrows, confirming the stability of ES approximations under deeper spike integration.

## B.4. Additional Operation Counts Comparisons

*Table 5.* QNN vs. SNN Function-Level Operation Counts Comparison (G=$10^9$)

| Function | Operator | QNN Count (G) | SNN (T=1) Count (G) | SNN (T=2) Count (G) | SNN (T=4) Count (G) |
|---|---|---|---|---|---|
| RMSNorm | MACs | 0.1051 | 0.0000 | 0.0000 | 0.0000 |
|  | ACs | 0.0524 | 0.0000 | 0.1049 | 0.3146 |
|  | Shifts | 0.0000 | 0.3145 | 0.3145 | 0.3145 |
| SiLU | MACs | 2.3953 | 0.0000 | 0.0000 | 0.0000 |
|  | ACs | 0.1409 | 0.1409 | 0.2818 | 0.5636 |
|  | Shifts | 0.0000 | 1.4090 | 2.8180 | 5.6361 |
| Softmax | MACs | 0.6554 | 0.0000 | 0.0000 | 0.0000 |
|  | ACs | 0.0406 | 0.4092 | 0.5321 | 0.7778 |
|  | Shifts | 0.0000 | 0.4096 | 0.4096 | 0.4096 |

To complement our analysis, Table 5 reports function-level operation counts. Our NLS-operators systematically replace multiply–accumulate (MAC) operations with accumulate (AC) and bit-shift operations, resulting in a more spike-friendly computation profile for neuromorphic hardware. The scaling behavior with time steps $T$ further reveals two implementation patterns. RMSNorm and Softmax perform their core shift-based computation only once, leading to constant shift counts while ACs grow linearly with $T$. In contrast, SiLU executes both shifts and ACs at every time step, causing the total operation count to scale linearly with $T$.

