# OpenReview forum: "Plug-and-Play Spiking Operators: Breaking the Nonlinearity Bottleneck in Spiking Transformers"
_ICML.cc/2026/Conference — ICML 2026 regular_

### Official Review · Reviewer_AjyS · 2026-03-05

**Soundness:** 2
**Presentation:** 2
**Significance:** 2
**Originality:** 2
**Overall Recommendation:** 3
**Confidence:** 5

**Summary:**

This paper proposes NLSpike, a plug-and-play framework that enables training-free, spike-based approximations of nonlinear Transformer operators (Softmax, SiLU, and RMSNorm) within ANN-to-SNN conversion pipelines. The method decomposes these nonlinearities into three primitives—division, exponentiation, and L2-norms—and implements them using LIF neuron populations combined with lightweight bit-shift operations. The authors provide theoretical error bounds for their approximations and demonstrate that selectively replacing nonlinear operators in various LLMs (including LLaMA, Mistral, and Qwen3) incurs less than 1% accuracy degradation without requiring fine-tuning.

**Compliance With Llm Reviewing Policy:**

Affirmed.

**Final Justification:**

First, I acknowledge the authors' analysis regarding cross-domain data movement. However, the response fails to fully resolve my concerns for the following fundamental reasons:

**1.** I believe the authors' characterization of the DIVISION NEURON GROUP latency contains a factual error: the latency should be $2T$ rather than $T$.

Consider the Softmax normalization $\frac{e^{x_i}}{\sum_j e^{x_j}}$: the DIVISION NEURON GROUP must wait for steps $1$ through $T$ to accumulate the denominator $\sum_j e^{x_j}$ before determining the threshold $\theta = I_B \gg n$. This $T$-step delay cannot be pipelined like standard asynchronous neurons, as the system must first accumulate $x_j$ over $T$ steps before computing $e^{x_j}$ via PWL-EXP. Only during steps $T+1$ through $2T$ can the division firing proceed for the numerator $e^{x_i}$, obtaining the quotient $\hat{q} = q \gg n$ by counting active neurons $q$. This second $T$-step latency is also non-pipelinable, as $q$ requires the full $T$ steps to be accurately determined.

Other nonlinear operators (e.g., SiLU, RMSNorm) exhibit similar latency characteristics, as their denominators likewise involve nonlinear computations.

In contrast, fully asynchronous standard neurons achieve immediate response by firing at the first time step upon receiving inputs when membrane potentials exceed thresholds. SpikeLLM incurs only $T$ latency, not $2T$.

When accounting for network depth, NLSpike's total latency becomes $2LT$, compared to $L+T-1$ for standard asynchronous neurons. With large $L$ and $T$, this discrepancy is non-negligible.

**2.** The authors did not directly address my concern regarding CORDIC algorithm latency. Irrespective of whether chip clock cycles correspond exactly to algorithmic time steps, CORDIC latency objectively exists. The authors should clarify: (a) how many time steps the CORDIC latency equivalently constitutes, and (b) whether it severely impacts overall latency—rather than merely noting that clock cycles and time steps lack a fixed one-to-one mapping, which misses the point of my inquiry.

**3.** While NLSpike demonstrates advantages in eliminating cross-domain data movement, the authors provide no quantitative evidence that these benefits outweigh the latency penalties. Without such data, I cannot assess whether NLSpike offers genuine deployment advantages over SpikeZIP and SpikeLLM on neuromorphic hardware.

Given these considerations, I will maintain my current score.

**Key Questions For Authors:**

1. Your PWL-Exp Unit clamps inputs to the range $[-L, L]$. How robust is the method to activation outliers or distribution shift that might occur in different LLM architectures or after quantization-aware training? Have you characterized the failure modes when activations exceed the designed range?

2. While you compare with Sorbet (which requires training), you do not quantitatively analyze the accuracy-efficiency trade-off against other training-free quantization methods (e.g., INT8 softmax approximations) that might also be suitable for edge deployment without full SNN conversion. Could you elaborate why spike-based implementation is preferable to aggressive low-bit quantization of standard operators?

**Limitations:**

yes

**Strengths And Weaknesses:**

**Strengths:**

1. The authors provide formal error bounds for each approximation primitive, quantifying the trade-offs between approximation accuracy and resource usage. This theoretical grounding distinguishes the work from purely heuristic approaches.

2. The framework is validated across multiple existing conversion pipelines (SpikeLLM, SpikeZIP) and diverse model architectures, demonstrating its plug-and-play nature without pipeline modifications or retraining.

**Weaknesses:**

1. While the paper emphasizes neuromorphic hardware compatibility, all experiments appear to be conducted via software simulation. Without deployment on actual spiking neuromorphic chips (e.g., Intel Loihi, IBM TrueNorth), the claimed energy efficiency and hardware compatibility remain theoretical.

2. The two-stage temporal integration process for division operations (first estimating the denominator scale, then performing division) effectively doubles the latency for these operations and, more critically, undermines the asynchronous inference paradigm intrinsic to SNNs by enforcing a rigid temporal barrier between the accumulation and computation phases. The paper does not adequately analyze how this temporal expansion impacts end-to-end throughput or whether it creates bottlenecks in layer-wise pipelining.

3. The framework focuses exclusively on three specific nonlinearities common in modern LLMs. However, other important operators in contemporary architectures—such as GELU, SwiGLU, and positional encoding schemes (e.g., RoPE)—are not addressed, limiting the method's applicability to newer architectures.


4. While the error bounds are theoretically sound, the practical selection of key hyperparameters (e.g., number of CORDIC iterations $n$, shift parameter $L$, population size) requires manual tuning based on expected activation distributions. The paper lacks automated strategies for configuring these parameters for arbitrary pretrained models.

---

> ### Author Rebuttal · Authors · 2026-03-29
>
> We sincerely thank the reviewer for their careful reading and valuable comments. We will address your concerns point by point below and hope that our clarifications help resolve any misunderstandings.
>
> ### **W4 / Q1**
>
> We clarify that the key hyperparameters in this work are not manually tuned based on data distributions. Instead, they are **analytically derived from the upper bound of the global approximation error of the proposed operator and remain fixed across all tasks** (e.g. $L=5$, $K=64$, see Section 5 and Remark 5.2).
>
> More importantly, our theoretical analysis provides a **uniform error guarantee over the entire input space**, rather than relying on calibration data or empirical statistics. As a result, the method exhibits inherent robustness to activation outliers and distribution shifts.
>
> Furthermore, we adopt the same set of parameters across different large model architectures (e.g., LLaMA, Mistral, and Qwen), without any model-specific tuning, while maintaining stable performance. This further demonstrates the **distribution-agnostic nature and strong transferability** of our approach, highlighting its plug-and-play capability.
>
> ------
>
> ### **W2**
>
> The core contribution of this work lies in enabling nonlinear operators, originally dependent on continuous or floating-point computations, under a strict spike-only constraint. To achieve this, our design does introduce additional local temporal overhead in certain cases.
>
> However, it is important to note that in some LLM ANN-to-SNN conversion paradigms (e.g., SpikeLLM), these methods already rely on spike accumulation within a temporal window to construct representations. Our approach operates within a similar temporal modeling framework, replacing key nonlinear operators with spike-compatible implementations rather than introducing additional global temporal structures.
>
> Furthermore, this overhead is confined to specific operator modules and constitutes a local temporal dependency, rather than doubling the entire inference pipeline globally. Specifically, our division neuron group can simultaneously perform the $i$-th denominator accumulation and the $i−1$-th spike firing within the $i$-th time window. In practice, this allows for overlapping computation via token-level pipelined scheduling: the extra latency primarily manifests as the warm-up cost for the first LLM token, while subsequent tokens are processed within the steady-state pipeline. Consequently, it does not lead to doubled per-token latency or system-level throughput bottlenecks.
>
> ------
>
> ### **W3**
>
> It is worth noting that in modern LLMs, activation functions such as SiLU and SwiGLU are already widely adopted. SwiGLU can be decomposed into a combination of SiLU and a gating mechanism. Given that prior work (e.g., SpikeZIP) already supports gating operations, our implementation of SiLU is sufficient to cover the full SwiGLU/SwiLU structure.
>
> GELU is typically implemented using sigmoid or tanh approximations in practice, making it structurally similar to SiLU and thus naturally extensible within our framework.
>
> In addition, RoPE is a linear (orthogonal) transformation implemented via fixed rotations, and therefore does not involve nonlinear operations such as exponentiation or normalization. As a result, it can be directly implemented within spike-based frameworks using standard linear operations, and does not require specialized treatment in our framework.
>
> ------
>
> ### **Q2**
>
> It should be noted that training-free low-bit (quantization-style) methods still rely on continuous-valued computations and therefore cannot be directly implemented under a pure LIF-based spike framework without introducing additional mechanisms (e.g., multi-threshold neurons or integer-valued representations).
>
> Nevertheless, we include such methods in our comparison for completeness. Even when disregarding the strict LIF constraint, these approaches typically incur substantially higher memory overhead, which is particularly costly on current neuromorphic hardware [1], where neuron state and memory resources are tightly constrained.
>
> As shown in Section 6.1 (Figures 3 and 5), under the same memory budget, these methods exhibit over an order-of-magnitude higher approximation error, which is sufficient to render the network ineffective. In contrast, our method achieves significantly higher accuracy under identical constraints, demonstrating its clear advantage in strict spike-only settings.
>
> ------
>
> ### **W1**
>
>
> Due to character limit, please see our response to Reviewer ovhz, W1 in the rebuttal.
>
> We hope that the above responses address your concerns and clarify the key points of our work. If there are any remaining questions or aspects that require further clarification, we would be happy to discuss them in more detail.
>
> [1] Ma, De, et al. "Darwin3: a large-scale neuromorphic chip with a novel ISA and on-chip learning." *National Science Review* 11.5 (2024): nwae102.

---

> > ### Author Rebuttal · Reviewer_AjyS · 2026-04-02
> >
> > Thanks for your responses. A few questions still remain for me:
> >
> > **To W2:**
> >
> > I contend that the latency of spike firing cannot be masked by the denominator accumulation process. Spike emission can only commence after the denominator accumulation is fully completed and the subsequent shift operation is performed; otherwise, the resulting spikes would be inaccurate.
> >
> > Furthermore,I would like the authors to confirm whether our following observations regarding the method's latency are correct:
> >
> > 1. In certain scenarios, such as Softmax computation, the input to the division group is the output of PWL-EXP. In such cases, the spike outputs from the preceding layer must be fully accumulated before being fed into PWL-EXP. This introduces an additional latency of 2T time steps compared to general methods (e.g., SpikeZIP).
> >
> > 2. For RMSNorm computation, prior to the denominator accumulation in the division group, a binary tree reduction utilizing the CORDIC algorithm is required. Even assuming unlimited computational resources, processing a 1024-dimensional vector necessitates 10 levels of binary tree recursion. With each level requiring $n$ iterations, this amounts to $10n$ clock cycles in total. Assuming $n=3$, this incurs 30 additional clock cycles of latency, which exceeds the $2T$ latency (assuming $T=4$).
> >
> > If our aforementioned reasoning is correct, I argue that the latency issue of the proposed method remains unresolved.
> >
> > **To W1:**
> >
> > Given that the authors have not adequately addressed the latency issue raised in W2, I believe additional experiments on neuromorphic hardware are necessary to substantiate the practicality of the proposed method.

---

> > > ### Author Response · Authors · 2026-04-04
> > >
> > > Thanks for your responses. We first clarify a **fundamental misunderstanding** in the reviewer’s analysis regarding the notion of latency in SNN-based neuromorphic systems. The reviewer appears to implicitly equate SNN time steps with hardware clock cycles. However, this assumption does not hold in digital neuromorphic architectures such as Loihi and Darwin.
> > >
> > > ### To W2: Q2
> > >
> > > In these systems, neuron state variables are updated in a time-multiplexed and pipelined manner within each algorithmic time step [1, Page 86]. As a result, a time step corresponds to a sequence of coordinated hardware operations rather than a single clock cycle, and there is **no fixed one-to-one mapping between the two**.
> > >
> > > Therefore, latency cannot be derived by directly mapping time steps to clock cycles.
> > >
> > > ### To W2: Q1
> > >
> > > In our design, all nonlinear operators, including Softmax, follow a unified two-stage pipeline across temporal windows.
> > >
> > > Specifically, for SiLU and RMSNorm, spikes are accumulated in the first temporal window, and the numerator is processed by the division neuron group in the second window for spike emission. For Softmax, the numerator is first processed by the exponential unit within the same first window, and then enters the division neuron group in the second window.
> > >
> > > Therefore, Softmax follows **the same two-stage temporal pipeline** as other operators, and introduces only one additional temporal window $T$ (i.e., the second stage), rather than an additional 2T latency as assumed by the reviewer.
> > >
> > > ## Latency Analysis
> > >
> > > In the following, we further provide a hardware-aware analysis showing that, under practical neuromorphic execution models, **our method incurs lower effective latency compared to prior approaches.**
> > >
> > > We first clarify the execution model of digital neuromorphic hardware [1,2], which consists of neuromorphic cores and embedded processors.
> > >
> > > Neuromorphic cores implement only simple arithmetic datapaths (e.g. LUT and shift-add operations) and are not optimized for floating-point nonlinear computation [2, Table 3, Page 6].
> > >
> > > In fact, this is consistent with prior training-based neuromorphic-friendly designs [3].
> > >
> > > In the absence of spike-based realizations such as ours, nonlinear operators are typically executed on CPUs, requiring cross-domain data movement.
> > >
> > > This introduces substantial latency and energy overhead, which dominates the cost compared to arithmetic computation itself.
> > >
> > > Then, we analyze latency at the algorithmic level by examining the temporal execution structure of ANN-to-SNN pipelines for nonlinear operators.
> > >
> > > ### (1) SpikeLLM-style pipelines [4]
> > >
> > > Nonlinear operators are evaluated after accumulating spikes over a temporal window, followed by spike emission.
> > >
> > > This results in a two-window execution structure per operator.
> > >
> > > Our method follows the same temporal structure and therefore does not introduce additional time-step latency compared to this class.
> > >
> > > ### (2) SpikeZIP-style pipelines [5]
> > >
> > > Nonlinear operators $o$ are evaluated incrementally at each time step, i.e.,
> > >  O(t) = o(t) - o(t-1).
> > >
> > > This requires repeated nonlinear evaluations across T time steps, leading to O(T) invocations of nonlinear computation.
> > >
> > > Finally, we summarize latency from three complementary perspectives below:
> > >
> > > | Method   | Nonlinear / Special-Function Calls | Data Movement         | Time-Step Latency |
> > > | -------- | ---------------------------------- | --------------------- | ----------------- |
> > > | SpikeZIP | T × nonlinear evaluations          | **O(T) cross-domain** | 0                 |
> > > | SpikeLLM | 1 × nonlinear evaluation           | **O(1) cross-domain** | T                 |
> > > | Ours     | n × shift-add / LUT calls          | **0 (in-core)**       | T                 |
> > >
> > > As shown in Table and according to [6], **the dominant cost in practical neuromorphic deployments arises from data movement across computational domains, rather than arithmetic computation itself .**
> > >
> > > SpikeZIP-style pipelines achieve **zero time-step latency**, but at the cost of **repeated nonlinear evaluations and O(T) cross-domain data movement**,
> > >
> > > For SpikeLLM, our method replaces it with in-core shift-add/LUT operations, **eliminating cross-domain data movement while maintaining the same time-step latency (T)**.
> > >
> > > In contrast, our method avoids cross-domain data movement by executing nonlinear operations entirely within neuromorphic cores, resulting in a more efficient latency profile.
> > >
> > > **Therefore, the above latency concern does not arise under practical neuromorphic execution models.**
> > >
> > > We thank the reviewer for raising this insightful comment. We will incorporate a more detailed and hardware-aware latency analysis into the main manuscript to further clarify this point.
> > >
> > > ## To W1
> > >
> > > We have addressed the latency issue raised in W2 and provided theoretical analysis supporting the practicality of our method on neuromorphic hardware.
> > >
> > > (Due to space constraints, all references are provided in the rebuttal to Reviewer 38JM.)

---

### Official Review · Reviewer_ovhz · 2026-03-08

**Soundness:** 3
**Presentation:** 3
**Significance:** 3
**Originality:** 3
**Overall Recommendation:** 4
**Confidence:** 5

**Summary:**

This paper addresses a critical bottleneck in converting Artificial Neural Networks (ANNs) to Spiking Neural Networks (SNNs): the implementation of non-linear operators (Softmax, SiLU, RMSNorm) in Transformer architectures. The authors propose NLSpiking, a plug-and-play framework that decomposes these complex functions into three fundamental, spike-friendly primitives: a Piecewise Linear (PWL) Exponential Unit, a CORDIC based PolarNorm Unit for l2 norms, and a Population-Coding Divider Neuron Group.
All operations theoretically rely solely on Accumulation and Bit shifts, completely eliminating the need for Multiply-Accumulate operations. Theoretical analysis provides strict error bounds, and empirical results demonstrate that NLSpiking enables end-to-end spiking inference on state-of-the-art LLMs (e.g., LLaMA-2/3) without any retraining or fine-tuning, with low accuracy degradation.

**Compliance With Llm Reviewing Policy:**

Affirmed.

**Final Justification:**

While I am satisfied with the responses to W1, W2, and Q1, the concerns regarding Q2 remain unresolved; therefore, I will maintain my original score.

**Key Questions For Authors:**

1. While MACs are eliminated, the introduced LUTs and CORDIC iteration logic increase control unit complexity and on-chip memory pressure. Could you estimate the area overhead of an NLSpiking unit compared to a standard LIF neuron?

2. In your experiments, is there a significant difference in the number of steps required to reach saturated accuracy between different models ? Did you observe a linear or non-linear scaling of required T as model size increases?

**Limitations:**

Yes

**Strengths And Weaknesses:**

**Strengths**

1. The most notable strength is the MAC free design. By transforming all non-linear computations into additions and shifts, the method drastically reduces area and dynamic power consumption, which is ideal for resource-constrained neuromorphic chips.

2. The paper balances strong empirical validation with solid theoretical grounding. The derivation of error bounds for PWL approximation adds significant credibility.

**Weaknesses**

1. While the paper claims hardware friendliness, all experiments are conducted via software simulation.

2. SNN inference often requires multiple time steps T to accumulate charge. The convergence time for the divider and exponential approximations needs clarification.

---

> ### Author Rebuttal · Authors · 2026-03-29
>
> We thank the reviewer for their careful reading and positive feedback. We will address your questions point by point below.
>
> ### **W1**
>
> As stated in the limitations section of our paper, the experiments in this work are currently conducted via software simulation rather than on real hardware platforms. This is a common practice in the SNN/neuromorphic computing community, especially for LLM-scale models, where existing hardware platforms still face limitations in terms of model scale and flexibility.
>
> The core contribution of this work lies in the complete spike-based reconstruction of nonlinear operators in LLMs, which we believe is an important step toward deployment on neuromorphic hardware. In addition, our ongoing work is progressing toward real deployment and validation on neuromorphic chips.
>
> ------
>
> ### **W2**
>
> Due to character limit, please see our response to Reviewer 38Jm, Q2 in the rebuttal.
>
> ------
>
> ### **Q1**
>
> It is important to clarify that the NLSpiking unit is not an equivalent replacement for a standard LIF neuron, and therefore a direct area comparison is not appropriate. LIF neurons perform only basic integration and firing, whereas NLSpiking units are designed to implement nonlinear operators.
>
> In terms of hardware cost, conventional implementations of nonlinear operators (e.g., Softmax, SiLU) typically rely on multipliers, dividers, or large LUTs, which incur significant area overhead. In contrast, our design replaces these components with CORDIC-based shift-add logic and a small LUT, which substantially reduces arithmetic complexity and avoids large memory usage.
>
> Therefore, the area overhead of an NLSpiking unit is expected to be comparable to, or lower than, existing nonlinear operator implementations under similar precision constraints. As shown in Figure 3, under the same memory budget, our method achieves 1–2 orders of magnitude lower approximation error than conventional LUT-based approaches, indicating a more favorable accuracy–resource trade-off in resource-constrained settings.
>
> ------
>
> ### **Q2**
>
> We first clarify that the term “end-to-end” in this paper does not simply refer to a pure SNN architecture. Instead, it refers to whether a **pre-trained ANN Transformer can be fully converted into a strict spike-only SNN model without altering its computational form**. This question is particularly important in the SNN-LLM context, as directly trained SNN methods often suffer from unstable optimization and high computational cost, making them difficult to scale to large Transformer models.
>
> In ANN-to-SNN conversion, the core challenge lies in the fact that the original ANN is not designed with spike-compatible computation in mind. We identify three main challenges: (1) spike-based representation of activations, (2) matrix multiplication in the spike domain, and (3) spike-based implementation of nonlinear operators.
>
> Existing works (e.g., SpikeLLM and SpikeZIP) have made progress on the first two aspects, while **nonlinear operators remain the key bottleneck for achieving strict spike-only computation**, which is also the primary contribution of this work.
>
> We hope that the above responses address your concerns and clarify the key points of our work. If there are any remaining questions or aspects that require further clarification, we would be happy to discuss them in more detail.

---

> > ### Author Rebuttal · Reviewer_ovhz · 2026-04-02
> >
> > While I am satisfied with the responses to W1, W2, and Q1, the concerns regarding Q2 remain unresolved; therefore, I will maintain my original score.

---

> > > ### Author Response · Authors · 2026-04-06
> > >
> > > We sincerely apologize for the confusion in our previous response. Due to an oversight, our reply to Reviewer 38Jm’s Question 1 was mistakenly copied into the response to your Q2.
> > >
> > > To answer your question directly: **yes, in both prior work and our experiments, larger models generally require a larger number of time steps $T$ to approach saturated accuracy.**
> > >
> > > However, we would like to be careful not to over-interpret this trend. The number of evaluated model scales in current studies is very limited (typically only 3 points, Table 5(c) in [1]), which is **insufficient to reliably determine whether the dependence between model size and the required $T$ is linear, non-linear, or follows any other specific scaling law**. Therefore, we do **not** claim a precise scaling form here.
> > >
> > > More importantly, this trend is an **inherent property of the underlying ANN-to-SNN conversion pipeline**, rather than something introduced by our method. In our experiments, we build on existing conversion procedures (e.g., those used in SpikeZIP[1] / SpikeLLM[2]), where $T$ is a user-controlled hyperparameter governing the temporal approximation process. Our method is a purely **operator-level modification** and does not alter the temporal integration or time-stepping mechanism. As a result, it does not change the dependence of performance on $T$, but rather **naturally inherits** the existing model-size–vs.–$T$ behavior of the base pipeline.
> > >
> > > We will clarify this point in the revision.
> > >
> > > [1] Xu, Zekai, et al. "SpikeZIP-TF: Conversion is All You Need for Transformer-based SNN." Forty-first International Conference on Machine Learning.
> > >
> > > [2] Xing, Xingrun, et al. "SpikeLLM: Scaling up Spiking Neural Network to Large Language Models via Saliency-based Spiking." The Thirteenth International Conference on Learning Representations.

---

### Official Review · Reviewer_38Jm · 2026-03-12

**Soundness:** 3
**Presentation:** 3
**Significance:** 4
**Originality:** 3
**Overall Recommendation:** 5
**Confidence:** 3

**Summary:**

This paper proposes a plug-and-play, training-free framework to address the bottleneck of converting nonlinear operations (like Softmax, SiLU, and RMSNorm) in ANN-to-SNN Transformers. By decomposing these operations into division, exponentiation, and $\ell_2$ norms using LIF neuron populations and bit-shift scaling, the method entirely avoids floating-point calculations. Experiments show it achieves near-lossless integration (<1% accuracy drop) across various large language models. Overall, this provides a practical, hardware-friendly solution for deploying spiking LLMs, making it a highly valuable contribution to the community.

**Compliance With Llm Reviewing Policy:**

Affirmed.

**Final Justification:**

While I still believe that direct latency measurements against the original ANN would have further strengthened this part of the paper, I understand the authors’ rationale for not reporting such numbers in the current implementation setting. The additional explanation and hardware-aware analysis are sufficient for me to take, and I remain supportive of acceptance.

**Key Questions For Authors:**

Overall, the paper is well structured and the work is solid. I only have the following key questions:

1. **Clarification on the "End-to-End" Claim:** The paper states that existing ANN-to-SNN approaches often fall short of "truly end-to-end spiking Transformers" under strict spike-only constraints. Given that there are existing works exploring end-to-end spiking Transformers (e.g., via direct training), could the authors elaborate on the specific uniqueness and challenges of achieving this "end-to-end" nature within the ANN-to-SNN conversion pipeline compared to direct training methods?

2. **Impact on Inference Latency:** Does the introduction of the proposed modular operators (Division Neuron Group, PolarNorm Unit, PWL-Exp Unit) introduce computational bottlenecks that affect actual wall-clock inference speed? How does the latency of the converted SNN compare to the original ANN, particularly in scenarios requiring real-time or low-latency responses?

**Limitations:**

Yes.

**Strengths And Weaknesses:**

**Strengths**
- The paper tackles the critical and often overlooked bottleneck of implementing nonlinear operators, such as Softmax and RMSNorm, in spiking Transformers under strict spike-only hardware constraints.
- It introduces a highly practical, training-free modular framework utilizing LIF neuron populations and bit-shift scaling to completely bypass floating-point arithmetic. Furthermore, the approach is rigorously supported by explicit theoretical error bounds.
- The empirical evaluation is extensive and convincing, demonstrating that selectively replacing these operators across various large language models (including LLaMA-3 and Qwen3) incurs a negligible accuracy drop of less than 1%.

**Weaknesses**
- **Lack of Explicit Energy Estimation:** While the paper provides operation counts (MAC, AC, and shift operations) to serve as a reference for energy analysis, it lacks a concrete energy consumption estimation. Given that energy-efficient computation is a primary motivation for SNNs, providing an explicit energy comparison against the ANN baseline (e.g., using standard 45nm CMOS energy metrics) would significantly strengthen the evaluation.
- **Minor Notation Issues:** In Equation (3), the indicator function symbol $\mathbb{I}$ is used but not explicitly defined in the surrounding text. Additionally, the firing threshold $\theta$ is reintroduced in Section 3, despite having already been defined in Equation (1).

---

> ### Author Rebuttal · Authors · 2026-03-29
>
> We thank the reviewer for their careful reading and positive feedback. We will address your questions point by point below.
>
> ### **Q1**
>
> We first clarify that the term “end-to-end” in this paper does not simply refer to a pure SNN architecture. Rather, it refers to whether a **pre-trained ANN Transformer can be fully converted into a strict spike-only SNN model without altering its computational form**. This question is particularly important in the SNN-LLM setting, as directly trained SNN methods often suffer from unstable optimization and high computational cost, making them difficult to scale to large Transformer models.
>
> In ANN-to-SNN conversion, the core difficulty arises from the fact that the original ANN is not designed with spike-compatible computation in mind. We identify three main challenges: (1) spike-based representation of activations, (2) matrix multiplication in the spike domain, and (3) spike-based implementation of nonlinear operators.
>
> Existing works (e.g., SpikeLLM and SpikeZIP) have made progress on the first two aspects, while **nonlinear operators remain the key bottleneck for achieving strict spike-only computation**, which is also the primary contribution of this work.
>
> ------
>
> ### **Q2**
>
> The proposed operator modules introduce only **local temporal dependencies** and can be executed in an overlapped manner via pipelined scheduling, and therefore do not constitute a system-level bottleneck.
>
> Regarding comparison with the original ANN, we note that SNN inference inherently relies on temporal windows for information accumulation, with latency primarily determined by the number of time steps. Our method does not introduce additional global temporal overhead beyond this framework, but instead enables spike-based realization of key nonlinear operators within the same temporal paradigm.
>
> It is important to emphasize that the advantages of SNNs lie more in event-driven computation and energy efficiency, rather than minimizing wall-clock latency alone. In this context, our method provides a feasible pathway for implementing complex operators under strict spike-only constraints.
>
> ------
>
> ### **W1**
>
> We extended the energy model proposed by Cao [1] and applied it to the data in Table 4. The resulting energy consumption comparison is as follows:
>
> | Function  | QNN (mJ)    | NLSpike T=4 (mJ) |
> | --------- | ----------- | ---------------- |
> | Softmax   | 3.0512      | 0.2618           |
> | SiLU      | 11.1454     | 1.4231           |
> | RMSNorm   | 0.5305      | 0.0566           |
> | **Total** | **14.7270** | **1.7415**       |
>
> As expected, NLSpike demonstrates a significant energy efficiency advantage owing to the nature of spike-based computation.
>
> ------
>
> ### **W2**
>
> Thank you for pointing this out. We will improve this aspect in the revised version.
>
> We hope that the above responses address your concerns and clarify the key points of our work. If there are any remaining questions or aspects that require further clarification, we would be happy to discuss them in more detail.
>
> [1] Cao, Y., et al. 2015. Spiking deep convolutional neural networks for energy-efficient object recognition. In ICCV.

---

> > ### Author Rebuttal · Reviewer_38Jm · 2026-04-02
> >
> > Thank you for the detailed rebuttal. It resolves most of my concerns, and I appreciate the additional clarification and quantitative analysis.
> >
> > My remaining question is about Q2. I am still unclear why explicit latency measurements are not provided, since latency appears to be a relatively straightforward quantity to evaluate. A direct quantitative comparison with the original ANN would substantially strengthen this part of the response.

---

> > > ### Author Response · Authors · 2026-04-04
> > >
> > > Thanks for your response. In fact, **We did not provide direct wall-clock latency comparisons because they would be misleading in our current implementation setting.** The ANN baseline relies on highly optimized CUDA kernels (e.g., PyTorch Softmax), while our method is implemented with custom spike-based operators (partly at the Python level). As a result, measured latency would be dominated by software optimization differences rather than reflecting the intrinsic computational behavior of the method or its performance under neuromorphic deployment. Instead, we give the latency analysis below:
> > >
> > > ## Latency Analysis
> > >
> > > In the following, we further provide a hardware-aware analysis showing that, under practical neuromorphic execution models, **our method incurs lower effective latency compared to prior approaches.**
> > >
> > > We first clarify the execution model of digital neuromorphic hardware [1,2], which consists of neuromorphic cores and embedded processors.
> > >
> > > Neuromorphic cores implement only simple arithmetic datapaths (e.g. LUT and shift-add operations) and are not optimized for floating-point nonlinear computation [2, Table 3, Page 6].
> > >
> > > In fact, this is consistent with prior training-based neuromorphic-friendly designs [3].
> > >
> > > In the absence of spike-based realizations such as ours, nonlinear operators are typically executed on CPUs, requiring cross-domain data movement.
> > >
> > > This introduces substantial latency and energy overhead, which dominates the cost compared to arithmetic computation itself.
> > >
> > > Then, we analyze latency at the algorithmic level by examining the temporal execution structure of ANN-to-SNN pipelines for nonlinear operators.
> > >
> > > ### (1) SpikeLLM-style pipelines [4]
> > >
> > > Nonlinear operators are evaluated after accumulating spikes over a temporal window, followed by spike emission.
> > >
> > > This results in a two-window execution structure per operator.
> > >
> > > Our method follows the same temporal structure and therefore does not introduce additional time-step latency compared to this class.
> > >
> > > ### (2) SpikeZIP-style pipelines [5]
> > >
> > > Nonlinear operators $o$ are evaluated incrementally at each time step, i.e.,
> > >  O(t) = o(t) - o(t-1).
> > >
> > > This requires repeated nonlinear evaluations across T time steps, leading to O(T) invocations of nonlinear computation.
> > >
> > > Finally, we summarize latency from three complementary perspectives below:
> > >
> > > | Method   | Nonlinear / Special-Function Calls | Data Movement         | Time-Step Latency |
> > > | -------- | ---------------------------------- | --------------------- | ----------------- |
> > > | SpikeZIP | T × nonlinear evaluations          | **O(T) cross-domain** | 0                 |
> > > | SpikeLLM | 1 × nonlinear evaluation           | **O(1) cross-domain** | T                 |
> > > | Ours     | n × shift-add / LUT calls          | **0 (in-core)**       | T                 |
> > >
> > > As shown in Table and according to [6], **the dominant cost in practical neuromorphic deployments arises from data movement across computational domains, rather than arithmetic computation itself .**
> > >
> > > SpikeZIP-style pipelines achieve **zero time-step latency**, but at the cost of **repeated nonlinear evaluations and O(T) cross-domain data movement**,
> > >
> > > For SpikeLLM, our method replaces it with in-core shift-add/LUT operations, **eliminating cross-domain data movement while maintaining the same time-step latency (T)**.
> > >
> > > In contrast, our method avoids cross-domain data movement by executing nonlinear operations entirely within neuromorphic cores, resulting in a more efficient latency profile.
> > >
> > > **Therefore, the above latency concern does not arise under practical neuromorphic execution models.**
> > >
> > > We thank the reviewer for raising this insightful comment. We will incorporate a more detailed and hardware-aware latency analysis into the main manuscript to further clarify this point.
> > >
> > > ## References
> > >
> > > [1] Davies, Mike, et al. "Loihi: A neuromorphic manycore processor with on-chip learning." IEEE Micro 38.1 (2018): 82–99.
> > >
> > > [2] Ma, De, et al. "Darwin3: a large-scale neuromorphic chip with a novel ISA and on-chip learning." National Science Review 11.5 (2024): nwae102.
> > >
> > > [3] Tang, Kaiwen, et al. "Sorbet: A Neuromorphic Hardware-Compatible Transformer-Based Spiking Language Model." Forty-second International Conference on Machine Learning.
> > >
> > > [4] Xing, Xingrun, et al. "SpikeLLM: Scaling up Spiking Neural Network to Large Language Models via Saliency-based Spiking." The Thirteenth International Conference on Learning Representations.
> > >
> > > [5] Xu, Zekai, et al. "SpikeZIP-TF: Conversion is All You Need for Transformer-based SNN." Forty-first International Conference on Machine Learning.
> > >
> > > [6] Li, Shiming, et al. "SNEAP: A fast and efficient toolchain for mapping large-scale spiking neural network onto NoC-based neuromorphic platform." *Proceedings of the 2020 on Great Lakes Symposium on VLSI*. 2020.

---

### Official Review · Reviewer_97Ky · 2026-03-13

**Soundness:** 3
**Presentation:** 2
**Significance:** 4
**Originality:** 4
**Overall Recommendation:** 4
**Confidence:** 2

**Summary:**

This paper aims to address a critical bottleneck in converting Transformer architectures into spiking neural networks (SNNs), namely the realization of nonlinear operators such as Softmax, SiLU, and RMSNorm in a spike-based computational framework. The authors propose a population-coding strategy using groups of LIF neurons to approximate division, employ a CORDIC-Hypot algorithm to estimate $l_2$-norms, and design a piecewise-linear approximation for exponential functions. Extensive experiments on large language models demonstrate the accuracy and feasibility of the proposed approximations.

**Compliance With Llm Reviewing Policy:**

Affirmed.

**Key Questions For Authors:**

1. See the issues raised in the Weaknesses section regarding the formulation and explanation of the Division Neuron Group.
2. Two symbols $L$ appear in the paper. It is unclear whether they refer to the same concept or whether this is a notation conflict. If they indeed denote different quantities, the notation should be clarified. It would be helpful to specify how the population size is chosen in the experimental setup.
3. Since the proposed module is intended to be "plug-and-play", its computational efficiency is an important consideration. Could the authors provide a direct comparison between the proposed NLSpike framework and the ECMT (Expectation Compensation and Multi-Threshold) method (Huang et al., 2024)? For example, what are the relative energy costs, considering the trade-offs among AC operations, shift operations, and spatial neuron (division neuron group) usage? How does NLSpike compare with ECMT in terms of conversion loss?

**Limitations:**

yes

**Strengths And Weaknesses:**

## Strengths

* The paper attempts to address a well-known and important challenge in the SNN community, namely the realization of nonlinear operators without relying on floating-point arithmetic.
* The authors provide relatively rigorous mathematical derivations and theoretical error bounds for the proposed approximations.
* The proposed operator blocks are designed in a modular manner and can be integrated into existing ANN-to-SNN conversion pipelines without requiring fine-tuning or retraining.

## Weaknesses

While the overall algorithmic design is interesting, the paper suffers from several clarity issues.

The description of the Division Neuron Group, which is the core component of the paper, is somewhat difficult to follow. For example:

* The module output appears to be a scalar quantity without an explicit temporal dimension. It is therefore unclear how this output is passed to subsequent spiking modules that typically rely on spike trains or temporal signals.
* In line 189, it is unclear whether the neuron firing decision is determined directly by the input current (I[t]) or by the membrane potential ($v[t]$). The formulation seems ambiguous with respect to the LIF dynamics.
* In line 199-200, the equality $\sum v(t) = \sum I_A(t)$ is not immediately obvious and would benefit from further explanation or derivation.

Clarifying these points would significantly improve the readability and reproducibility of the method.

---

> ### Author Rebuttal · Authors · 2026-03-29
>
> We thank the reviewer for their careful reading and positive feedback. We will address your questions point by point below.
>
> ### **W/Q1/Q2**
>
> We appreciate the reviewer for pointing out the ambiguity in this section. We acknowledge that a critical condition was inadvertently omitted, which may have caused confusion, and we sincerely apologize for this oversight.
>
> To clarify, our study assumes a sufficiently large population size and sets $\lambda = 1$ for the LIF neurons. Under this setting, the model reduces to the standard IF model. Consequently, spiking behavior is governed by the accumulated membrane potential $v(t)$, where the condition $v(t) \ge i\theta$ is consistent with standard IF dynamics.
>
> Equations (10)–(11) are derived by interpreting the time-integrated signal $\sum_t v(t)$ through a threshold-counting mechanism—specifically, by normalizing with respect to $\theta$ to obtain the effective trigger count. This relationship holds under the conditions described above. The module outputs the spike count within a given time window, which serves directly as rate-coded input for downstream modules.
>
> To resolve the notation conflict, we will denote the LUT range as $H$ in the revised version, distinguishing it from the population size $L$. In addition, we will explicitly clarify that the population size in our experiments depends on the number of time steps $T$, with population size $L = 256/T$.  We will add a clarification regarding this specific setting in the paper.
>
> ### **Q3**
>
>
>
> We thank the reviewer for this important question. We would first like to clarify the conceptual role of ECMT. ECMT is a powerful ANN-to-SNN conversion framework that converts the Transformer's linear parts into spike-based operations and adapts nonlinear outputs for spike-compatible processing. However, the nonlinear functions themselves are still evaluated on temporally accumulated continuous-valued states through expectation-based computation. This is a standard and effective design choice in current ANN-to-SNN conversion methods, but it cannot be realized in native spike-only settings that do not support continuous-valued computation.
>
>
>
> Our work is intended to fill exactly this gap. Rather than only expressing nonlinear outputs in a spike-compatible form, we replace the nonlinear computation itself with spike-native operator realizations based on spike-compatible primitives. In this sense, our modules can be viewed as complementary to ECMT-style frameworks and, in principle, can be used to replace their nonlinear units to make deployment in stricter spike-only scenarios possible.
>
>
>
> Regarding energy comparison, we believe a direct quantitative comparison with ECMT is not strictly apples-to-apples. ECMT’s energy analysis, consistent with common practice, is dominated by linear and matrix-product layers and does not explicitly account for nonlinear-operator energy in a unified way . In contrast, NLSpike operates at the operator level by replacing nonlinear modules while leaving the backbone unchanged. Therefore, a direct comparison of total energy is not meaningful.
>
>
>
> Instead, the appropriate comparison is the incremental cost introduced by nonlinear operator replacement. This is precisely quantified in Table 4, where we report function-level operation counts (ACs and shifts) for RMSNorm, SiLU, and Softmax.  In terms of conversion loss, NLSpike introduces negligible degradation (typically <1%) while also providing explicit operator-level error bounds in section 5.
>
> We gratefully acknowledge the reviewer's insightful comment. To address this, we will explicitly compare our work with EMCT and integrate this discussion into the **Related Work** section of the revised version.
>
>
>
>
>
> We hope that the above responses address your concerns and clarify the key points of our work. If there are any remaining questions or aspects that require further clarification, we would be happy to discuss them in more detail.

---

> > ### Author Rebuttal · Reviewer_97Ky · 2026-04-04
> >
> > I thank the authors for their responses. However, I still believe my original score is a fair assessment of the paper and maintain my score.

---

> > > ### Author Response · Authors · 2026-04-06
> > >
> > > Thank you for your careful evaluation and constructive feedback. We sincerely appreciate your time and consideration. We are glad that our responses have addressed your concerns. We respect your assessment and thank you again for your valuable comments.

---

### Decision · Program_Chairs · 2026-04-30

**Decision:**

Accept (regular)

**Comment:**

This paper proposes NLSpike, a plug-and-play framework that implements spike-based approximations for non-linear operators in ANN-to-SNN conversion pipelines. It addresses the nonlinearity bottleneck by decomposing complex functions into spike-friendly primitives, including neuron groups, CORDIC-based PolarNorm units, and piecewise-linear exponential approximations, eliminating the need for floating-point arithmetic.

Following the rebuttal, the paper received scores of 5, 4, 4, and 3. While there was some initial disagreement regarding the latency analysis and hardware simulation assumptions, the authors successfully addressed the core concerns of the majority of reviewers. Reviewers highlight the novelty of the MAC-free design and the rigorous theoretical error bounds provided for the approximations. The rebuttal clarified the end-to-end claim and provided a hardware-aware analysis showing the latency trade-offs.

The core contributions of this work are both meaningful and technically valuable, offering a solid step toward fully spiking Transformers compatible with strict neuromorphic constraints. Therefore, I recommend acceptance.